# Quantum sensing for gravity cartography

Ben Stray[1,6], Andrew Lamb[1,6], Aisha Kaushik[1], Jamie Vovrosh[1], Anthony Rodgers[1], Jonathan Winch[1], Farzad Hayati[1], Daniel Boddice[2], Artur Stabrawa[1], Alexander Niggebaum[1], Mehdi Langlois[1], Yu-Hung Lien[1], Samuel Lellouch[1], Sanaz Roshanmanesh[2], Kevin Ridley[1], Geoffrey de Villiers[1], Gareth Brown[3], Trevor Cross[4], George Tuckwell[2,5], Asaad Faramarzi[2], Nicole Metje[2], Kai Bongs[1] & Michael Holynski[1✉]

The sensing of gravity has emerged as a tool in geophysics applications such as engineering and climate research[1–3], including the monitoring of temporal variations in aquifers[4] and geodesy[5]. However, it is impractical to use gravity cartography to resolve metre-scale underground features because of the long measurement times needed for the removal of vibrational noise[6]. Here we overcome this limitation by realizing a practical quantum gravity gradient sensor. Our design suppresses the effects of micro-seismic and laser noise, thermal and magnetic field variations, and instrument tilt. The instrument achieves a statistical uncertainty of 20 E (1 E = $10^{-9}$ s$^{-2}$) and is used to perform a 0.5-metre-spatial-resolution survey across an 8.5-metre-long line, detecting a 2-metre tunnel with a signal-to-noise ratio of 8. Using a Bayesian inference method, we determine the centre to ±0.19 metres horizontally and the centre depth as (1.89 −0.59/+2.3) metres. The removal of vibrational noise enables improvements in instrument performance to directly translate into reduced measurement time in mapping. The sensor parameters are compatible with applications in mapping aquifers and evaluating impacts on the water table[7], archaeology[8–11], determination of soil properties[12] and water content[13], and reducing the risk of unforeseen ground conditions in the construction of critical energy, transport and utilities infrastructure[14], providing a new window into the underground.

The quantum gravity gradient sensor uses atom interferometry[15], which has been used in laboratory-based experiments to provide sensitive measurements of gravity[16], to investigate the equivalence principle[17], the fine-structure constant[18] and Newton's gravitational constant[19], prompting the desire to transition these sensors into practical devices for use in real-world environments[20]. For example, gravity sensors have been created that can be used on volcanoes and mountain environments[21,22], and for measurements by air[23], by sea[24] and on rockets[25]. A typical approach in these devices is to use light pulses to drive two-photon stimulated Raman transitions in atoms and use these to create a superposition of matter waves in different momentum and energy states. The resulting atomic wavepackets move along two spatially separated trajectories, before being recombined and interfered. This creates the matter-wave analogue of a Mach–Zehnder interferometer. The phase difference in the resulting interference pattern is proportional to the local gravitational field. However, such devices, as with any gravimeter, are fundamentally limited in their measurement time owing to the need to average out micro-seismic vibration[26]. This presents a major barrier to realizing gravity maps with high spatial resolution.

To enable gravity cartography, and operation in application-relevant conditions, we implement an 'hourglass' configuration cold atom gravity gradiometer[27]. This enables robust coupled differential measurements on two clouds of atoms, separated by a vertical baseline[28]. Two counter-oriented single-beam magneto-optical traps (MOTs) allow passage of common Raman beams to perform interferometry (Fig. 1a). The measurement axis is aligned to measure the vertical component, $G_{zz}$, of the (3 × 3) gravity gradient tensor, which is the largest and most relevant component for gravity cartography. Differential operation suppresses primary noise sources (vibration and micro-seismic), systematic shifts (such as tilt) and changes in the optical path length between the beams used to drive the Raman transitions[29]. The commonality of laser intensity noise for the cooling beams of the single-beam MOTs[6] enables cloud temperature fluctuations to be stable to within a few hundred nanokelvins (Fig. 1b, top panel), limiting the impact of a.c. Stark shifts, and reduces cloud centre-of-mass motion by an order of magnitude when compared to conventional six-beam approaches (see Methods). The resulting changes in baseline are below 75 ppm (Fig. 1b, bottom panel), which corresponds to a systematic error of less than 0.1 E.

The hourglass configuration provides several practical benefits (see Methods). Avoiding the need for off-axis beams creates a robust and compact optical delivery arrangement, allowing months of operation in the field with no need to correct alignment. The configuration also provides a radially compact form factor, enabling compact magnetic shielding with 25 dB attenuation that suppresses effects due to external magnetic fields, preventing these from affecting the atom cloud generation. The beam configuration, in combination with a robust all-fibre laser system, enables independent control of the counter-propagating Raman beams, facilitating reversal of the

[1]Midlands Ultracold Atom Research Centre, School of Physics and Astronomy, University of Birmingham, Birmingham, UK. [2]School of Engineering, University of Birmingham, Birmingham, UK. [3]Dstl, Porton Down, Salisbury, UK. [4]Teledyne e2v, Chelmsford, UK. [5]RSK, Hemel Hempstead, UK. [6]These authors contributed equally: Ben Stray, Andrew Lamb. ✉e-mail: m.holynski@bham.ac.uk

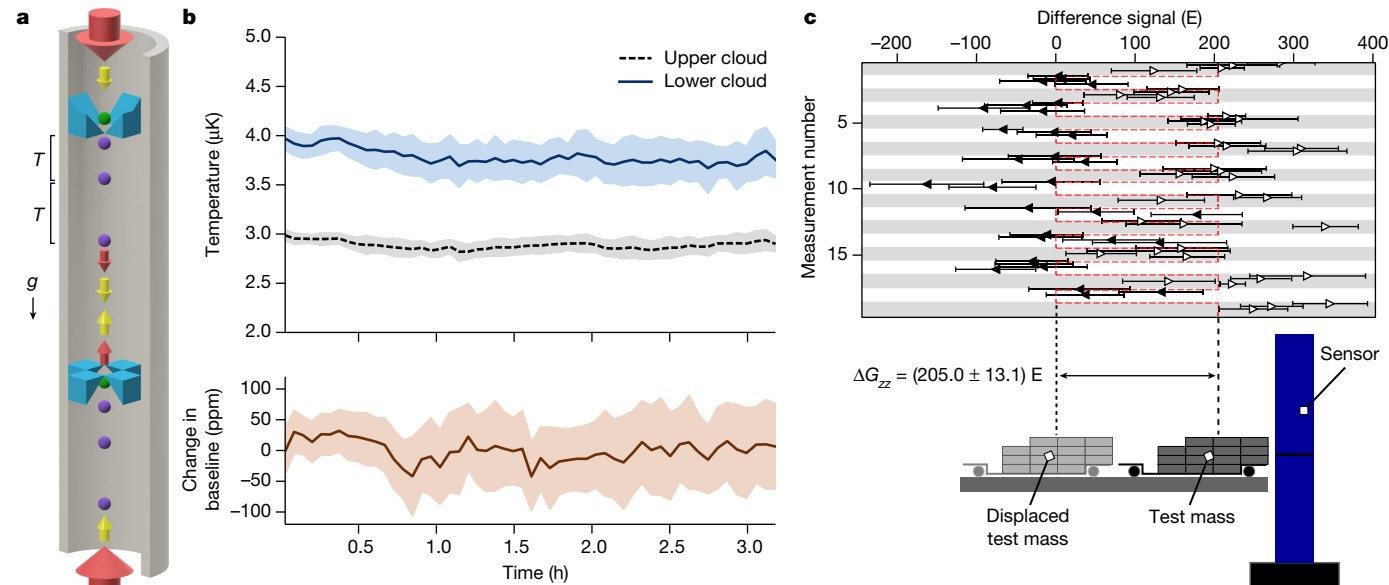

**Fig. 1 | Hourglass gradiometer. a**, Hourglass gradiometer using two counter-oriented single-beam MOTs, realized using mirror assemblies (blue). The initial atom clouds (green) fall under local gravitational acceleration, $g$, before being subjected to light pulses separated by time $T$ to realize the atom interferometers (purple). The beam delivery is indicated with arrows (see Methods for details). The cooling beams (red) are deflected by the in-vacuum mirrors (blue) to provide cooling in all directions, with the central portion of each input beam passing through the aperture between the mirrors to provide the final cooling beam for the opposite MOT. The atom interferometry beams (yellow arrows) have a smaller beam waist, such that they pass through the mirror aperture without significant clipping. Each interferometer is operated simultaneously, with a vertical baseline separation of 1 m. **b**, Temporal variation of atom cloud temperatures from each trapping region (top panel), measured using time of flight[41], and the relative change of the 1 m cloud separation baseline over time (bottom panel) (solid lines: averaged data at a bin size of 50 measurements at 4 s per measurement; shaded regions: σ range of the averaged data), determined from time of arrival. **c**, Measurement of the gravity gradient variation caused by movement of a test mass between two positions— either close to the sensor (open points) or displaced from the sensor (filled points). Each measurement number represents a specific position of the test mass, with the odd measurement numbers having the mass close to the sensor. Each data point is formed from the average of eight gravity gradient measurements, with each of those containing 25 shots from the atom interferometer each taking 1.5 s. The error bar for each data point is the standard error of the eight gravity gradient readings. The test mass was moved approximately every 20 min, with a variation of ±3.5 min, and its position was repeatable to approximately 1 cm. The modelled projection of the change in gravity gradient signal, $\Delta G_{zz}$, is shown in red.

light-pulse directions[30]. Interleaving measurements in each direction suppresses several systematic effects, including reducing those due to residual magnetic fields to below measurement precision. Furthermore, phase shifts and contrast loss from parasitic Raman transitions[31] are prevented through independent delivery of the Raman beams for each direction, without the need for a phase lock.

To measure the gravity gradient (see Methods), each MOT is loaded for 1 to 1.5 s with $^{87}$Rb atoms before sub-Doppler cooling is used to reduce the cloud temperatures to the microkelvin regime. The clouds are then dropped and simultaneously subjected to an atom interferometry sequence. The output of each interferometer is measured using fluorescence to detect the ratio of the populations of the two relevant atomic ground states, with approximately $10^5$ atoms participating in each atom interferometer, for a typical measurement rate of 0.7 Hz. A Lissajous plot of the upper versus lower atom interferometer outputs is then used to extract the differential phase, from which the gravity gradient is determined (Fig. 2, inset)[32]. The sensor was verified under laboratory conditions by modulating the position of known test masses near the sensor to vary $G_{zz}$ (Fig. 1c). This resulted in a measured change of (205 ± 13.1) E, compared to a modelled signal of 202 E.

Similarly, the sensitivity and stability of the instrument were evaluated in an outdoor environment. The Allan deviation[33] of the phase data (Fig. 2) showed an average short-term sensitivity of (466 ± 8) E/√Hz and a statistical uncertainty of 20 E within 10 min of measurement.

To demonstrate the potential for gravity cartography, a 0.5-m-spatial-resolution survey was performed along an 8.5-m survey line above a pre-existing multi-utility tunnel. This is a tunnel with a 2-m by 2-m internal cross-section and a reinforced concrete wall of approximately 0.2-m thickness. It is situated underneath a road surface that is located between two multistorey buildings. Nearby buildings and terrain around the survey site provide further signals[34] that can mask targets of interest. To estimate the expected signal from the tunnel, a model of

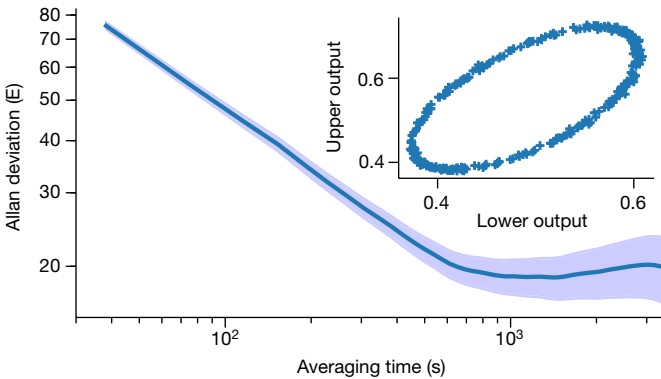

**Fig. 2 | Gradiometer performance.** Allan deviation, with overlapping averages, of the instrument output during outdoor operation over approximately 8.6 h, shown with percentage error. Inset: a typical subset of the ellipse data (300 points) used for the Allan deviation, showing a Lissajous figure of the output signal of the upper and lower interferometers, which is used to extract the gradiometric phase. The deviation of the ellipse from a circle gives rise to a clustering of points around the extremal points of the ellipse.

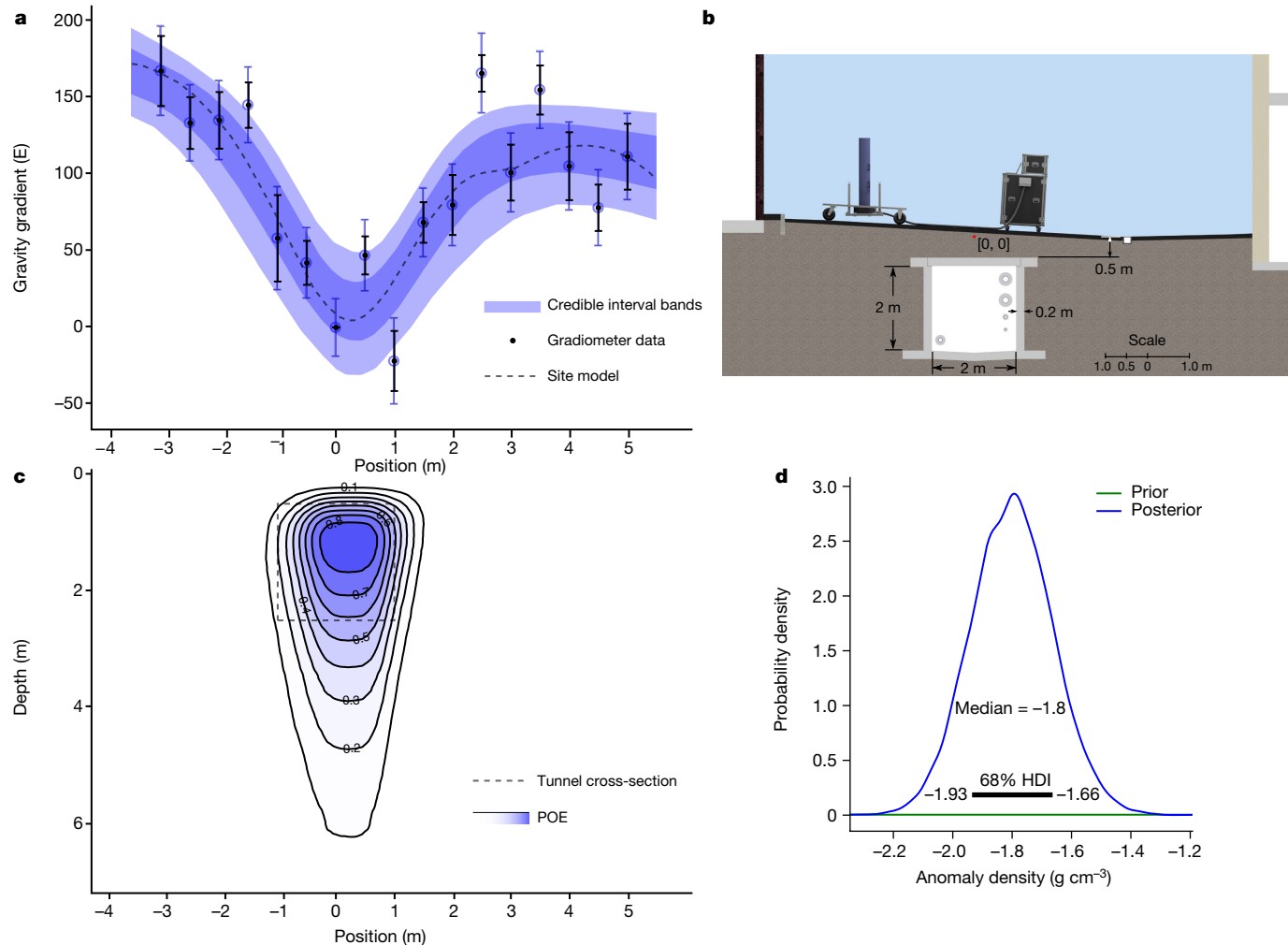

**Fig. 3 | Field survey. a**, Gravity gradient data with standard error (black scatter) and total inferred uncertainty when including model uncertainty (blue scatter), 1$\sigma$ (dark blue shading) and 2$\sigma$ (light blue shading) credible interval bands, and a site model (dashed) (see Methods). **b**, Scale schematic of the site showing the dimensions and position of the tunnel and leading contributions to the gravity signal. The origin for coordinates (red dot) is defined in the vertical direction by the lowest point on the survey line, and in the horizontal direction by the expected location of the centre of the tunnel. **c**, POE (blue contours) inferred from the gravity gradiometer data, and expected tunnel position (dashed). **d**, Estimation of soil density obtained by shifting the focus of the inference process to use assumed knowledge of the tunnel geometry and inference of the gradiometer data, showing the 68% highest density interval (HDI).

the site was constructed using an air/soil contrast infinite cuboid void, taking into account local buildings and terrain. The parameters for the model were informed using building plans (CAD files), with these being cross-checked using ground-penetrating radar, and auxiliary data from on-site measurements such as topography scanning. This provided an estimated peak signal from the tunnel of 150 E, which corresponds to a phase change of 17.5 mrad for the atom interferometer. Fig. 3a shows a comparison between the site model and the atom interferometer data, showing that the measurement data are consistent with what is expected for a gravity gradient anomaly with the expected location and size of the tunnel. A scale representation of the site and tunnel, including local buildings and site topology, is shown in Fig. 3b.

For use in practical applications, it will be important to interpret the data in an accessible way that produces information on which a user can make decisions or act. For this purpose, we have developed a Bayesian inference method and applied this to the gradiometer data with a data-generated model of a buried cuboid[35] assumed a priori. This uses the gradiometer data in conjunction with estimates of the site and geophysical parameters (as detailed in Extended Data Table 2) to make quantitative predictions of the depth and spatial extent of the anomaly.

For instance, we assume that the soil density is within the expected range for the type of soil at the survey site by using a Gaussian distribution, with a mean of −1.80 g cm$^{-3}$, to represent a void in surrounding soil, and standard deviation of 0.10 g cm$^{-3}$. The inference process produces distributions for the position, depth and cross-sectional area of the tunnel using the probability of excavation (POE) metric[36] (Fig. 3c). The observed spread of the POE is expected, due to measurement uncertainty and the ambiguity that exists between model parameters specifying depth, area and density, typical of inference from potential field data[37]. A signal-to-noise ratio of 8 for the detection is estimated from the data, finding the deduced horizontal position of the tunnel centre at (0.19 ± 0.19) m along the survey line and a depth to the centre of (1.89 −0.59/+2.3) m (see Methods).

Furthermore, by assuming a priori knowledge of the tunnel geometry and including topographical information of the survey site, the focus of the inference was switched to infer the soil density (Fig. 3d). This results in a near-Gaussian posterior distribution for the density parameter, with a mean of −1.80 g cm$^{-3}$ and standard deviation of 0.15 g cm$^{-3}$.

The statistical uncertainty demonstrated by the prototype instrument during static operation (which, for the 20 E gradiometer

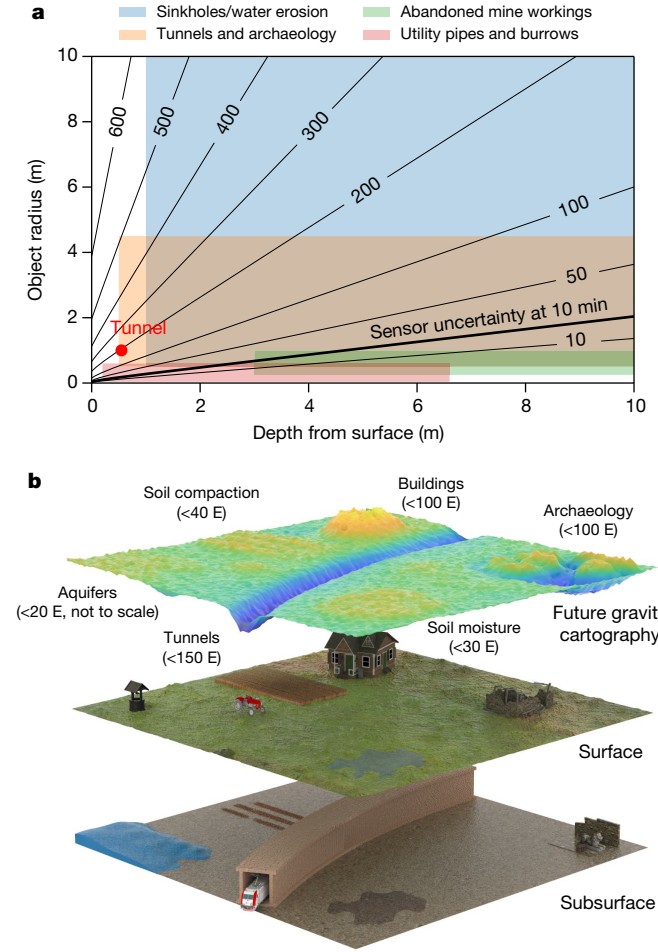

**Fig. 4 | Application relevance. a**, Contour plot for typical gravity gradient signal sizes in various applications, contours in eötvös for density contrasts of 1.8 g cm⁻³. The parameters of the detected feature (red dot) and the statistical uncertainty of the sensor (solid line) are shown. Features in the region above the solid line are detectable with current uncertainty. **b**, A perspective of future gravity cartography being used with 0.5 m spatial resolution over a region, at an uncertainty level of 20 E. Expected signal sizes for a range of applications are shown.

uncertainty, is equivalent to a 1.4 ng uncertainty for each of its two gravimeters) surpasses the reported performance of commercial gravimeters for survey applications by a factor of 1.5–4 (ref. [38]). In this first demonstration of submetre-resolution mapping with quantum gravity sensors, the repeatability of the prototype during the survey was similar to that of commercial gravimeters and limited by systematic effects (see Methods), such as due to the Coriolis effect, which can be addressed through further engineering. Furthermore, the sensor could be moved from one spatial position to another within 75 s, including alignment to the vertical to within 1 millidegree. If addressing these aspects, such as through operation on a rail or vehicle, the current instrument performance would in principle allow detection of the tunnel, or similar anomaly, with a 10-point line scan and a signal-to-noise ratio of 3 within 15 min of total measurement time.

The detection of the tunnel allows the assessment of the instrument performance for a range of potential applications. Fig. 4a shows a range of typical signal sizes for a variety of application areas in comparison to the statistical uncertainty of the sensor, with features in the range above this being detectable with the current instrument. In civil engineering applications, this performance could provide a reduction in the uncertainty of ground conditions and be used to inspect brownfield sites, to search for tunnels and large or near-surface utilities, and to detect erosion features before they become sinkholes. This performance is also relevant to archaeological applications (for example, enabling the detection of tombs or hidden chambers and investigating how previous civilizations used underground infrastructure). Furthermore, the sensor could be of particular use in the mapping of aquifers, to better understand and optimize the use of water and its impact on the environment. It could also be used to measure density distributions within the ground. On the basis of the inferred standard deviation, the soil density extraction method is currently sensitive to 10% level changes in the mean, meaning that in principle this could distinguish between soil that is either dry or saturated, or used to investigate localized soil compaction (for example, in precision agriculture). Typical anticipated signals for these applications, with the 20 E statistical uncertainty that our sensor achieves within a 10 min measurement time, are illustrated in Fig. 4b.

The removal of vibration noise means, in contrast to the case in gravimeters, that future improvements in instrument sensitivity can be directly translated into reductions in measurement time or improved uncertainty. Implementation of further scientific enhancements to the sensor, including, for example, the use of large-momentum beamsplitters[39,40], has the potential to provide a further 10- to 100-fold improvement in instrument sensitivity, allowing faster mapping or detection of smaller and deeper features. It is expected that such performance will be achieved in practical instruments within the next 5–10 years.

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

## Methods

### Experimental overview

A light-pulse atom interferometer is conceptually similar to an optical interferometer, with the roles of light and matter interchanged. Atoms, acting as matter waves, are subjected to a sequence of light pulses that impart momentum to them, acting analogously to mirrors and beam-splitters. Applying a light pulse for an appropriate length of time will cause a transition between the ground and excited states of an atom, accompanied by the absorption and stimulated emission of a photon. Such a pulse, commonly referred to as a π pulse, acts as an atom optic mirror owing to the momentum that is transferred. Similarly, tuning the light pulse such that it has only a 50% transition probability, commonly referred to as a π/2 pulse, acts as a beamsplitter through providing a momentum kick to only half of the atomic probability distribution. A matter-wave equivalent of the optical Mach–Zehnder interferometer can then be created through applying a π/2–π–π/2 pulse sequence with an evolution time, $T$, between the pulses. The resulting space-time area enclosed by the atomic trajectories (Extended Data Fig. 1b) is proportional to the local gravitational acceleration, which can then be measured from the relative population of the two atomic states after the final pulse.

A gravity gradiometer utilizes two such interferometers offset vertically and probed simultaneously with the same pulse sequence. This suppresses common-mode effects, such as noise from vibration or phase changes due to variations in tilt with respect to the gravity of the Earth, which are indistinguishable from the gravity anomalies of interest according to Einstein's equivalence principle. Our device consists of two subunits (Extended Data Fig. 1a), a sensor head and a control system, with light and electrical signals transferred through a 5-m umbilical. The gradiometer is shown in Extended Data Fig. 2, with an overview of its size, weight and power characteristics.

The sensor head features a vacuum system with dual MOT preparation and interrogation regions in an hourglass configuration, with all light delivered to the atoms through on-axis counter-oriented telescopes. The light is delivered in each direction, with portions of the beam being redirected towards the atom-trapping region using in-vacuum mirrors, to form the radial cooling beams in each MOT. The central portion passes through, such that each input provides the vertical laser cooling beam in a given direction for both MOTs. This makes all fluctuations in intensity common for the radial cooling beams (preventing lateral offsets), and, through use of a Gaussian beam shape, provides a higher intensity for the vertical beams to better saturate the radiation pressure force in this direction. This results in a greatly improved stability and robustness of the laser cooling process, reducing fluctuations in temperature or atom cloud position (Fig. 1b) without the need for excessive laser powers that would inhibit field operation. In a comparable test system, this provided a reduction in average cloud centre-of-mass motion to $(0.14 \pm 0.09)$ mm as compared to $(1.19 \pm 0.86)$ mm over an hour in similar conditions with a six-beam MOT. Both MOT regions have two coils, each formed of 92 turns of 1-mm-Kapton-coated copper wire wound around an aluminium former (fixed using epoxy), with a slit to prevent eddy currents. The coils have a radius of 43 mm and separation of 56 mm, to produce a linear field gradient of 12.5 G cm$^{-1}$ at a driving current of 2.5 A. These are located around the vacuum system, such that the strong magnetic field axis of their quadrupole field is along the direction of travel of the cooling beam axis. In addition, two sets of rectangular coil pairs, each having 20 turns, are located around the MOT regions. These have a separation of 100 mm, and dimensions of 320 mm in the vertical and 90 mm in the horizontal, and can be used to compensate residual magnetic fields, or apply offsets. In practice, no compensation fields are used for the molasses phase. In the lower chamber, one coil pair is used to apply a 0.63 G field to adjust the atom cloud horizontal position by approximately 0.5 mm in the MOT phase, improving the interferometer

contrast. A bias coil[42] is positioned around the system to define a quantization axis and remove degeneracy between magnetic sublevels, with other coils being switched off after the magneto-optical trapping phase. This has a variable pitch shape to account for edge effects and improve field uniformity over the atom interferometry region. The system is enclosed in a magnetic shield that provides 25 dB attenuation of the external field. The in situ magnetic field profile is measured (through spectroscopy of the Raman transition) as being homogeneous to below 5% across the atom interferometry region, limited by internal magnetic field sources from vacuum pumps.

The laser system consists of telecom lasers that are frequency doubled to 780 nm, to be near the D2 line of rubidium-87 (refs. [43,44]). The light for laser cooling is generated by passing the laser output through an electro-optic modulator (EOM) and generating a sideband at a frequency of approximately 1.2 GHz output from the carrier. This is used to provide a locking signal using the $|F = 3> \rightarrow |F' = 4>$ transition in rubidium-85, placing the carrier frequency such that it is tuneable around resonance with the $|F = 2> \rightarrow |F' = 3>$ transition of rubidium-87 to provide the cooling light. A separate EOM is used to provide repumping light resonant with the $|F = 1> \rightarrow |F' = 2>$ transition. Atom interferometry is realized through two-photon stimulated Raman transitions. The Raman laser used to drive these has a linewidth of 73 kHz and is locked with an offset of 1.9 GHz to the $|F = 2> \rightarrow |F' = 3>$ transition. The second Raman frequency is generated using a pair of EOMs operating at 6.835 GHz. Performing the differential measurement suppresses phase noise that may arise owing to optical path-length changes between the two Raman beams (such as those due to vibration and thermally induced changes in the refractive index of fibres). This allows the two beams to be delivered independently without the need for a phase lock between them, facilitating an implementation in which the modulated spectrum is applied to only one of the input beams. This avoids parasitic Raman transitions that give rise to systematic offsets and dephasing when using conventional modulation-based schemes, such as those including a retro-reflected beam[31]. To realize a practical implementation of space-time area reversal[30], also known as wavevector reversal, the system has an EOM in each input direction of the Raman beams, and the modulation signal is applied to one arm in each measurement. This allows the direction of the momentum kick imparted to the atoms to be changed between measurements, by changing which arm the modulation signal is applied to using a radiofrequency switch (see Extended Data Fig. 1). The contributions to the interferometer phases due to acceleration under gravity are sensitive to the direction of the recoil imparted by the light, whereas those arising from many other effects, such as those due to magnetic fields, are not. This allows these effects to be removed when interleaved measurements are performed in the two recoil directions.

The light is delivered to the sensor head using polarization-maintaining optical fibres, with separate fibres for the cooling and Raman beams. These fibres deliver the light to optical telescopes that collimate the light at the desired beam size. The cooling beams have a waist of 24 mm, and contain a typical maximum power of 130 mW. These impinge on the in-vacuum mirrors, which are 15-mm right-angle prisms (Thorlabs, MRA15-E03), to deliver the horizontal cooling beams. The mirrors are mounted to a titanium structure (attached using Epo-Tek H21D adhesive) in a cross configuration such that there is a 15-mm aperture in their centre. The central portion of the cooling beams passes through these apertures to provide the sixth beam required for the opposite MOT. The Raman beams are overlapped with the cooling beams using a polarizing beamsplitter cube, such that they are then delivered along the same beam axis as the cooling light. The Raman beams, each containing a typical maximum power of 300 mW, have their waist set to 6.2 mm to limit aperturing and diffraction on the central aperture of the in-vacuum mirrors, allowing the Raman beams to pass through the system without being redirected by the prisms. Although aperturing is limited on the mirrors in the current instrument, it may be desirable to

use a larger Raman beam than the aperture in more compact systems or those aiming to further reduce dephasing induced by laser beam inhomogeneity. Diffraction from the aperture would need to be given due consideration if pursuing this, as would the potential for further light shifts due to, in this case, one interferometer seeing extra light fields from mirror reflections. The polarization of the light is set to the appropriate configuration for cooling or driving Raman transitions through use of voltage-controlled variable retarder plates in the upper and lower telescopes used to deliver the light. The intensity of the Raman beams is actively stabilized using feedback from a photodiode to control acousto-optic modulators, which are also used to produce the laser pulses.

The experimental sequence starts by collecting approximately $10^8$ rubidium-87 atoms in each MOT from a background vapour over 1–1.5 s. Molasses cooling is then used to reduce the upper- and lower-cloud temperatures to $(2.86 \pm 0.09)$ μK and $(3.70 \pm 0.20)$ μK, respectively (see Fig. 1b). The differences in temperature arise from differences in local residual magnetic fields, arising primarily from the magnetic shield geometry, and small differences in optical alignment. Optical state and velocity selection is performed to select only atoms in the $|F = 1, m_F = 0\rangle$ magnetic sublevel and desired velocity class. This is achieved through application of π pulses and a series of blow-away pulses to remove atoms in undesired states and velocity classes. Atom interferometry is then performed with a pulse separation of $T = 85$ ms and π-pulse length of 4 μs. The interferometers are read out using bistate fluorescence detection to determine the atomic state population ratios of the $|F = 2\rangle$ and $|F = 1\rangle$ ground states, for which $(2.7 \pm 0.1) \times 10^5$ and $(1.7 \pm 0.1) \times 10^5$ atoms participate in the upper and lower interferometers, respectively, with a typical measurement rate of 0.7 Hz. The differential phase, from which the gravity gradient is derived, is extracted by plotting the upper interferometer outputs against the lower interferometer outputs, to form a Lissajous plot as shown in the inset of Fig. 2. In addition to random noise arising from vibration, we add a deliberate random phase value, from between 0 and 2π, to the final pulse of the interferometer. At ellipse phases that do not correspond to a circle, a clustering of points around the extremal points of the ellipse is visible even for uniform noise.

The quantum projection noise of the system based on the participating atom number is approximately 44 E/√Hz. The total noise budget includes contributions from further terms, and is shown in Extended Data Table 1, alongside relevant systematics observed during the survey. The noise budget was investigated through computer simulation of noise processes, compared to experimental data, and ellipse fitting.

### Survey practice and processing of the measurement data

For each measurement on the survey, 600 runs of the atom interferometer were typically taken with the sensor head in one location (with the horizontal position being measured using a total station, Leica TS15, and the vertical position from the road surface being approximately 0.5 m for the lower sensor and 1.5 m for the upper sensor), giving twelve 25-point ellipses in each of the interferometer directions and therefore 12 separate estimates of the gravity gradient. Repeat measurements were taken on each measurement position, with typically three points on each position. A measurement was taken at a base station between each measurement point, with the final base-station measurement for one location used as the first for the next. The quality of fitting to each ellipse was identified using the error metric, $\varepsilon$, defined as

$$\varepsilon = \frac{\left(\frac{1}{a} + \frac{1}{c}\right)}{2}\left(\frac{1}{N}\sum_{i=1}^{N} L_i^2\right)^{\frac{1}{2}},$$

in which $N$ is the number of data points, $L$ is the minimum distance between each data point and a point on the ellipse, and $a$ and $c$

correspond to an ellipse defined parametrically by equations $x = a\sin\theta + b$ and $y = c\sin(\theta + \phi) + d$, respectively. Errors in the ellipse fitting are sensitive to changes in the ellipse opening angle[47]. On the basis of numerical simulations, we estimate this effect to be less than a few parts in one thousand; therefore, a 100 E change would be subject to an error of less than 0.5 E. Such errors are therefore small compared to other errors. Such a 100 E change in gradient would correspond to an 11.6 mrad change in the ellipse shape. This phase shift can be compared to a 2π measurement range, meaning that measurement range of the instrument in this configuration is relevant to the majority of practical features of interest (these being typically below 400 E).

Ellipse fits found to have $\varepsilon > 0.05$ were automatically discarded. This resulted in 98.4% of all data being usable in normal operation, representing a favourable data up time compared to that of similar conventional geophysical devices.

To process the data, a straight line was fitted to the base-station points, with this line then being subtracted from all data points. This is standard practice to remove drift in geophysical surveys. The leading source of drift is believed to be due to the a.c. Stark shift, with this also being relevant owing to the difference in the temperature of the two clouds. The gravity gradient value is then taken as the average of the measurement points, resulting in an estimate of the difference in gradient between the measurement location and the base station. Furthermore, the variations in the data points are used to make an estimate of the error in the difference value. When multiple measurements from the same location were combined, a weighted average was used, giving less weight to measurements with greater errors. The weighting factor is proportional to the reciprocal of the variance of each measurement[48]. The data, as shown in Fig. 3a, are not corrected for terrain or effects such as tides. Tidal effects are not corrected, being negligible through the differential measurement of the gravity gradient.

The average of the gravity gradient error found across the measurement positions of the survey is 17.9 E. Comparing this to an approximate signal size of 150 E gives an approximate signal-to-noise ratio of 8.

### Inference from gravity gradiometer data

Bayesian inference is a framework within which prior beliefs can be updated with information contained in data. For a model parameter vector ($\theta$) and a data vector ($d$)

$$p(\theta|d) = \frac{p(d|\theta)p(\theta)}{p(d)},$$

in which $p(d|\theta)$ is the likelihood, $p(\theta)$ is the prior, $p(d)$ is a normalization constant and $p(\theta|d)$ is the posterior distribution.

The likelihood function provides the misfit between the measured data, $d$, and the modelled data values calculated from the model parameter vector, $\theta$. The model used here is that of a three-dimensional cuboid[35]; the free model parameters are shown in Extended Data Fig. 3, along with the functional form of the respective prior distributions. The rationale behind the chosen prior distributions is detailed in Extended Data Table 2. The total uncertainty for each measurement point is calculated using the Pythagorean sum of the standard error and the model uncertainty random variable multiplied by the average of the standard error across all of the measurement positions.

The probabilistic Python package pymc3 (ref. [49]) is used to implement the cuboid model, define the model parameter prior distributions and sample the posterior distribution, using a no U-turn sampler[50]. Extended Data Fig. 4 shows the Bayesian posterior distribution for select model parameters.

The parameter posterior distributions represent the updated beliefs about the model parameters, given the measurement data. To aid interpretation of the posterior distribution, the POE[36] is calculated, which represents the spatial probability of the anomaly underground, given the model and prior distributions (as shown in Fig. 3c). The

horizontal position of the tunnel centre is determined as (0.19 ± 0.19) m along the survey line, with the distribution being approximately Gaussian. The depth from the origin, defined in the vertical using the lowest point on the survey line, to the centre is (1.7 −0.59/+2.3) m. At the horizontal position of the tunnel, the distance to the surface from the origin is approximately 0.19 m, meaning that the total distance from the surface to the tunnel centre is (1.89 −0.59/+2.3) m. From the tunnel geometry, this places the top of the tunnel at approximately 0.89 m depth from the surface.

The signals arising from local features are used to create a distinct site model. This is used to provide an estimate of the expected shape of the gravity gradient signal over the site, for comparison with the inference output. These features include the tunnel of interest, basements from nearby buildings, walls and a drain. They are shown in the scale drawing of Fig. 3b.

## Data availability

The datasets generated and/or analysed during the current study are available on an open data repository. This is located at https://doi.org/10.25500/edata.bham.00000740. Source data are provided with this paper.

## Code availability

The code that supports the findings of this study is available on an open repository. This is located at https://doi.org/10.25500/edata.bham.00000740.

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

**Acknowledgements** This work was supported by the EPSRC (EP/M013294/1 and EP/T001046/1) and Innovate UK (104613), and the gradiometer was built under a contract with the Ministry of Defence, as part of the UK National Quantum Technologies Programme. We acknowledge S. Bennett for input on the manuscript and activity, and technical support from the UoB EPS workshop.

**Author contributions** The design and development of the sensor was performed by B.S., A.L., A.K., J.V., A.R., J.W., F.H., K.B. and M.H. with inputs from A.S., A.N., K.R., G.d.V., M.L., Y.-H.L. and S.L., and inputs on design for field use from D.B., N.M., T.C., G.T. and G.B. The characterization and calibration measurements were provided by B.S., J.V., J.W., F.H., K.R. and M.H. The survey site modelling was performed by D.B., A.L., A.R., K.R. and F.H. with inputs from N.M. and G.T. The survey design and measurements were contributed by J.V., D.B., J.W., F.H., K.R., S.R., B.S., A.L. and M.H. with input on the survey design and process from N.M., K.B., G.T. and A.F. Data processing was carried out by J.W., K.R., J.V. and A.R., with A.R. providing the Bayesian inference. M.H. and K.B. conceived and coordinated the experiment. M.H., B.S., A.L., K.B., J.V. and A.R. wrote the manuscript. All authors contributed to the review and improvement of the manuscript.

**Competing interests** The University of Birmingham has filed a patent application based on the gradiometer design, with M.H., A.L., G.d.V. and K.B. listed as inventors (number 20200386906 16/772517). G.T. is employed by a company that make commercial use of gravity sensing. T.C. is employed by a company involved in the development of quantum technology and is a member of the UK Quantum Technology Strategic Advisory Board. G.B. is employed by DSTL, which is connected to the Ministry of Defence, who are a funder of the work. The authors declare no other competing interests.

**Additional information**
**Correspondence and requests for materials** should be addressed to Michael Holynski.

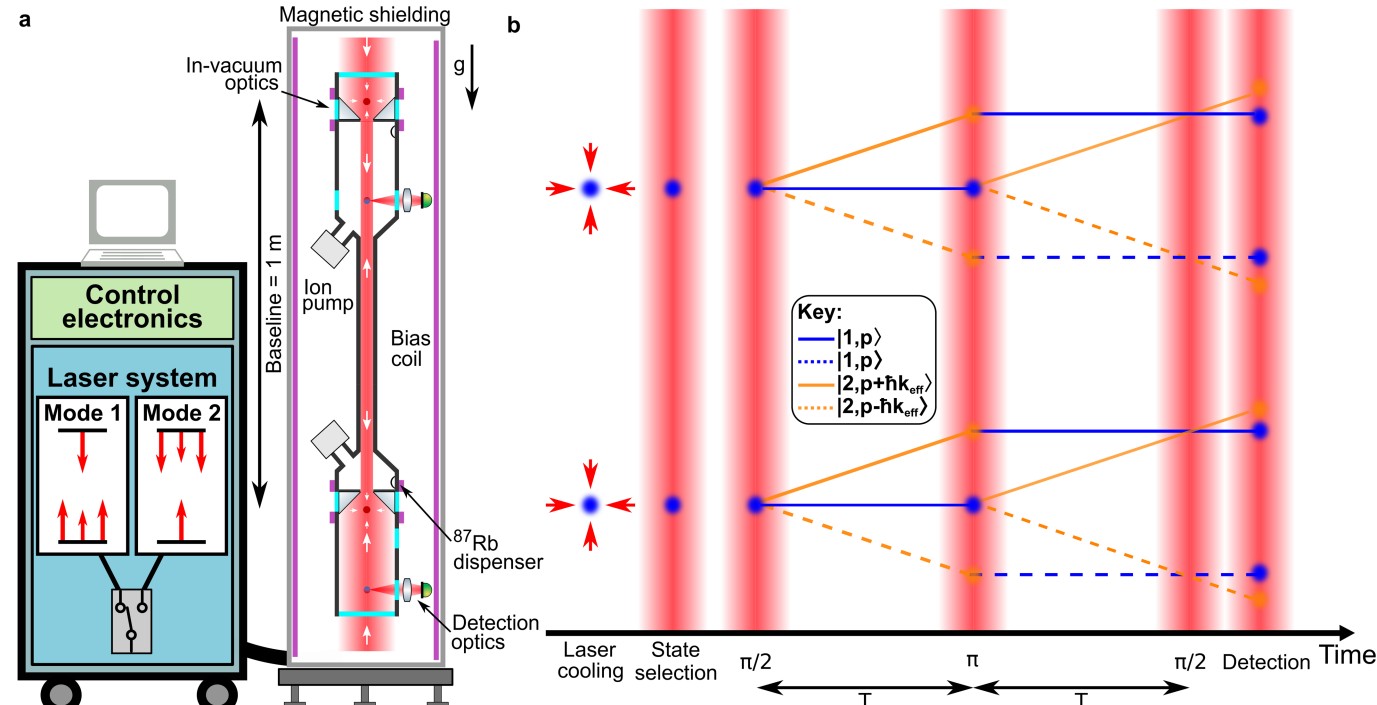

**Extended Data Fig. 1 | Overview of the experimental system and sequence.**
The system is formed of the main sensor head and an enclosure for the laser and control systems, with the laser system showing the two modes of Raman beam delivery that are used, with arrows representing the beams input to the chamber. The sensor head is formed using the hourglass configuration. This keeps all beam delivery along the central axis, improving stability and allowing the use of a radially compact magnetic shield. The laser system is formed of telecom lasers which are frequency doubled to 780 nm, to be near to the D2 transition line of rubidium-87. The laser light and electronic signals pass through an umbilical to reach the sensor head, with the laser light being delivered from the top and bottom of the sensor. The experimental sequence begins with atoms being loaded into two 3D MOTs, and then being dropped by turning off the laser light. While in free-fall, a sequence of velocity selective Raman pulses and blow-away pulses are used to select only the desired magnetic sub-level state and velocity class, with other atoms being removed from the sequence. This is followed by a π/2- π- π/2 interferometry sequence. The Raman transitions are realized using EOMs to create sidebands at a frequency difference equal to the hyperfine ground state splitting. In contrast to previous approaches[31,45,46], each input direction contains a separate EOM with the driving frequency being applied to only one input direction, such that the laser frequencies for the upward and downward Raman beams are in either mode 1 or mode 2 of the spectral configurations shown in the figure. This removes the effect of parasitic Raman transitions that create offsets and contrast loss in conventional modulation based approaches. The use of this laser scheme is enabled through the hourglass configuration allowing independent delivery of the Raman beams, while suppressing phase noise through differential operation. Switching between these two modes changes the input direction of the modulated beam spectrum, changing the direction of the first momentum kick in the interferometer and causing it to open in the opposite direction (dashed lines in the interferometer sequence). This allows a practical implementation of the wavevector reversal procedure[30], where the contributions to the phase due to the gravitational acceleration are sensitive to the direction of the recoil imparted by the light, while many other effects such as those due to magnetic fields are not. Interleaving measurements with interferometers running in each of these modes removes these sources of error while doubling the contribution due to gravity. Finally, the interferometer outputs are read out by measuring the atomic state populations of the two hyperfine ground states, using a fluorescence pulse delivered along the central axis, with the light that is scattered by the atoms being captured on a photodiode.

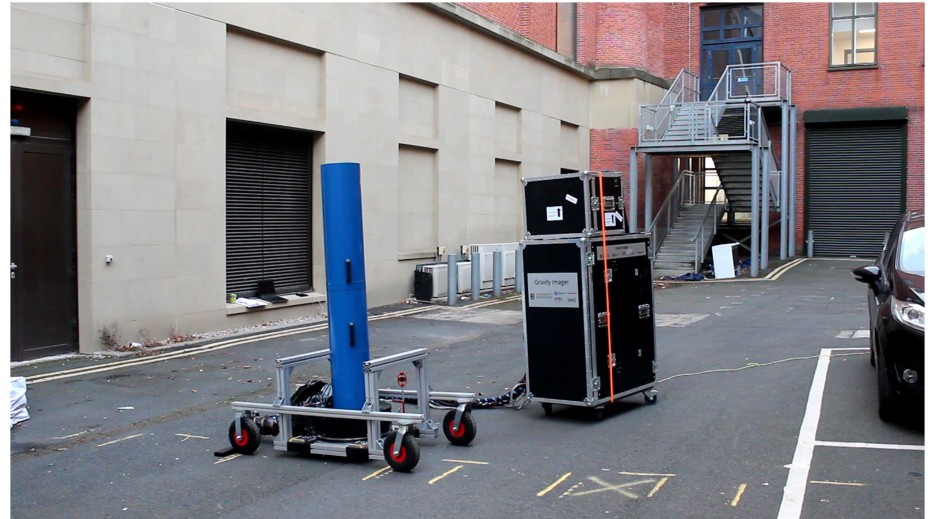

**Extended Data Fig. 2 | Gravity gradiometer on the survey line, above the tunnel.** The main sensor head (blue cylinder) stands at 1.87 m high, varying by 5 cm depending on the setting of the adjustable feet. This places the measurement positions of the sensor at approximately 0.5 m from the road surface for the lower sensor, and 1.5 m for the upper sensor. The floor footprint of the sensor head is 0.64 m by 0.6 m, and the upright cylinder has a diameter of 0.27 m. The total weight of the sensor head is approximately 75 kg. This is connected to a flight case, which contains the lasers and control system. This has an internal height of 24 U standard rack units (1 U = 4.4 cm), an external footprint of 1.10 m by 0.46 m and height of 1.34 m. A secondary case is placed on top of this, with dimensions of 0.50 m by 0.59 m by 0.46 m. The combined weight of these cases is approximately 250 kg. The system operates on a single mains wall socket, drawing approximately 800 W and having a short-term battery holder over. It can also operate on a generator supply without any observed additional noise. Also shown is a frame used to move the sensor head, and prism used to reference position with a total station.

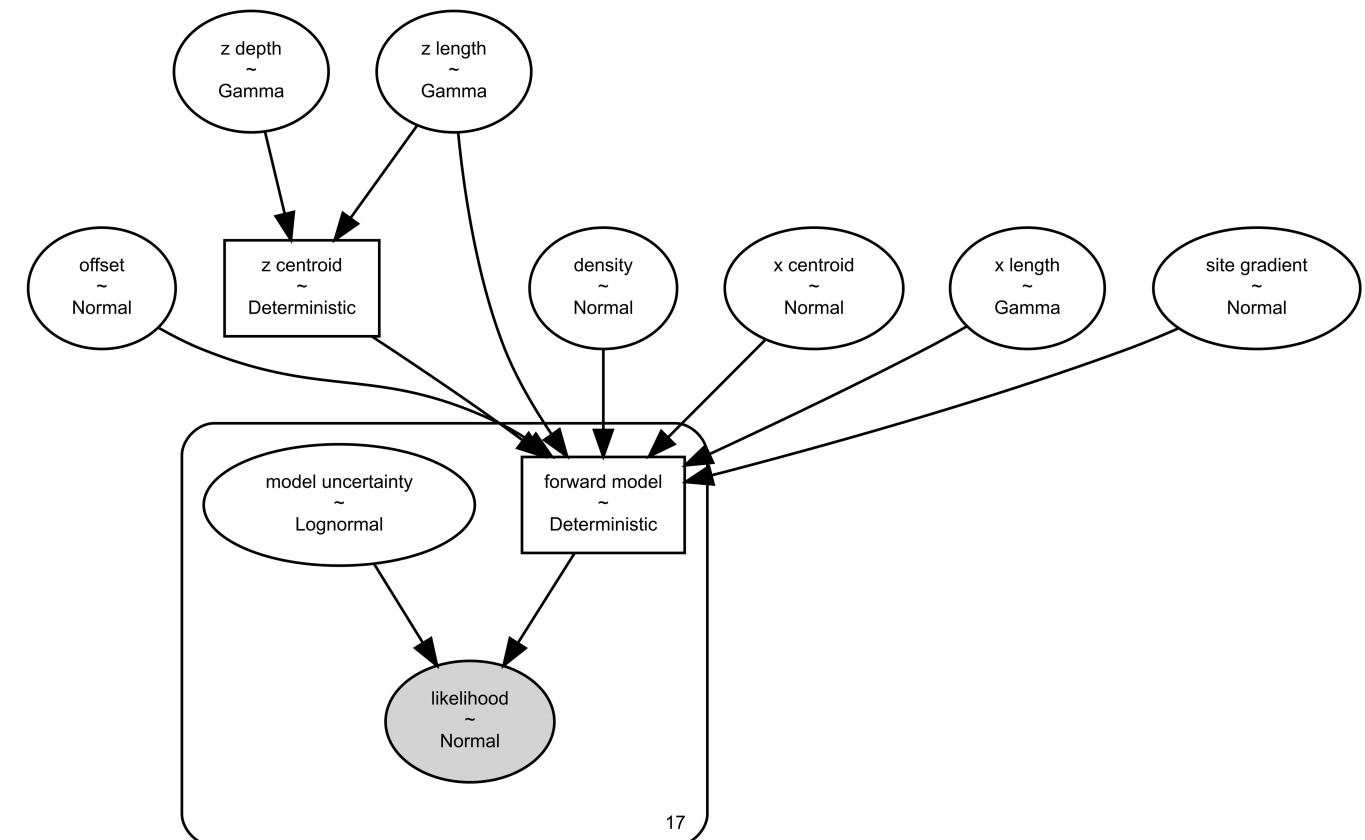

**Extended Data Fig. 3 | Relationship between model parameters (white ovals), with their respective prior distribution form, to the normal likelihood distribution (grey oval).** Deterministic parameters are shown in rectangular boxes. Parameters contained inside the rounded edge rectangle are all one dimensional arrays of length 17, the length of the gradiometer data set.

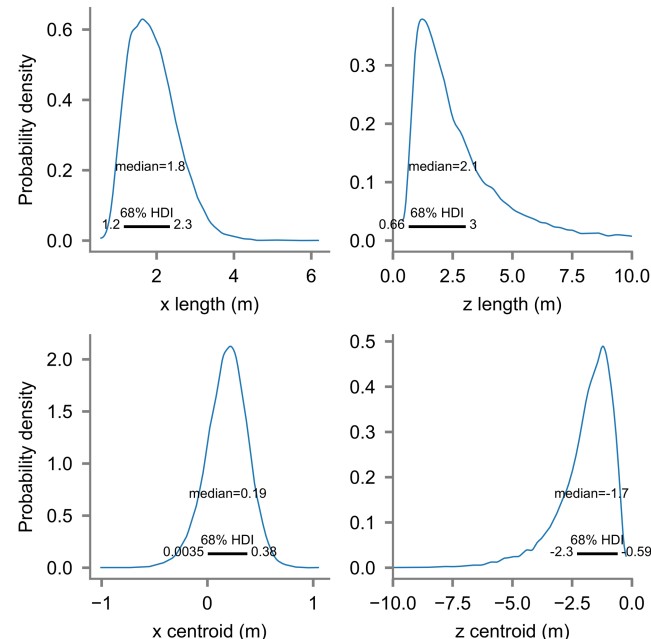

**Extended Data Fig. 4 | Bayesian posterior distributions for selected model parameters.** The horizontal and vertical directions are represented by x and z respectively. The median and 68% highest density interval (HDI, represented by the black lines and numerical extents) are shown for each distribution. The shape of the z length and z centroid distributions is non-Gaussian due to the known depth ambiguity for gravity sensing and asymmetries in the boundary condition, i.e. the parameters being limited by the ground surface above the tunnel.

## Extended Data Table 1 | Sensor noise budget and survey systematics

**Sensor noise budget**

| Uncertainty source | Limiting noise level | Comments |
|---|---|---|
| Photodiode thermal noise (Johnson noise) | 144 E/√Hz | Measured from detection photodiodes and read out chain with no light present |
| Background light noise | 344 E/√Hz | Measured using reference pulses with no atoms present. |
| Atom shot noise | 184 E/√Hz | Number of atoms measured from photodiode signals using known detection geometry and detector sensitivity. This exceeds simple estimates of quantum projection noise due to noise arising from those atoms that do not participate in the interferometer. This is due to fringe contrast being less than 100% and because the gradiometer extracts differential phase through an ellipse-fitting process in the presence of common-mode noise, rather than phase differencing the outputs of two ideally-balanced interferometers. |

**Survey systematics**

| Source | Systematic shift | Comments |
|---|---|---|
| Coriolis shifts due to orientation angle (yaw) error | < 26 E point to point | Calibrated during survey, finding 5.9 E/degree at orientation angle used in survey. Angle controlled to within 4.5 degrees, and measured using a total station (Leica TS15). |
| Changes in cloud separation baseline | < 0.1 E | Estimated via time of flight, Fig. 1b. |
| Projection error due to instrument tilt (pitch, roll) error | < 1 E point to point | Calibrated, measuring 1 E/millidegree shifts, representing 350x improvement over the projection shifts that would be observed for an ideal gravity sensor. Angle controlled to 0.001 degrees during survey using an inclinometer (Jewell Instruments DMP-2-10-232-AMP) and a machined pitch on the instrument feet. The inclinometer is used to continuously monitor tilt, and is permanently attached to a rigid base plate which also acts as the main interface between the instrument feet and the rest of the instrument. |
| Magnetic fields on site | <43 E across site | No shift observed across a calibration range of 221 mGauss (measured at the lower cloud height) vertical field, applied using a coil of diameter 0.85 m, placed at the base of the sensor. Shifts are below measurement resolution. It is possible alternative geometries create different shifts, with the shape of the calibration field approximating that over the survey site. The magnitude of the applied field matches the peak-to-peak variation across the site for a measurement at the sensor height. The standard error of the calibration measurement was 43 E. The magnetic field measurements for the site and calibration field were performed with a Bartington Grad-13. |

Contributions to the uncertainty budget of the sensor during the measurement of the data set for Fig. 2 are shown, with shifts during the survey of Fig. 3.

**Extended Data Table 2 | Prior distributions defined for inference model random variables**

| Model parameter (unit) | Explanation | Distribution | Rationale |
|---|---|---|---|
| x centroid (m) | Cuboid centroid in the x direction. | $\mathcal{N}(\mu = 0, \sigma = 4)$ | Cuboid unlikely to be located outside of the measurement line. |
| z depth (m) | Depth to the top of the cuboid. | $\gamma(\alpha = 2, \beta = 0.5)$ | Cuboid unlikely to be deeper than the measurement line width. |
| x length (m) | Length of the cuboid in the x direction. | $\gamma(\alpha = 2, \beta = 0.5)$ | Positive value, unlikely to be larger than the measurement line width. |
| z length (m) | Length of the cuboid in the z direction. | $\gamma(\alpha = 2, \beta = 0.5)$ | Positive value, unlikely to be deeper than the measurement line width. |
| Density (g/cm$^3$) | Density difference between the void and surrounding soil. | $\mathcal{N}(\mu = -1.8, \sigma = 0.1)$ | Informative prior, encoding feasible region of average soil density. |
| Site gradient (E/m) | Gravity gradient slope across the site. | $\mathcal{N}(\mu = 0, \sigma = 10)$ | Uninformative prior. |
| Offset (E) | Offset between the measured data and the modelled data. | $\mathcal{N}(\mu = 0, \sigma = 10)$ | Uninformative prior. |
| Model uncertainty (dimensionless) | The simple cuboid model has limited explanatory power to deal sufficiently with measurement outliers. | $\text{Log}(\mathcal{N}(\mu = 0, \sigma = 0.25))$; Shape = (17, 1) | Multiplied by the average uncertainty of the measurement points. Encoding that we expect an uncertainty of comparable magnitude to the average measurement uncertainty, due to the limited explanatory power of the simple cuboid forward model. |

Horizontal and vertical displacements are represented by x and z respectively. The normal and gamma distributions are shown, with their respective parameters.