## [Peer Review File · Nature]

Manuscript Title: Quantum sensing for gravity cartography

Reviewer Comments & Author Rebuttals

Reviewer Reports on the Initial Version:

Referee #1 (Remarks to the Author):

The authors demonstrated a mobile atomic gradiometer for detecting underground tunnels. The gradiometer is constructed with two vertically separated atom interferometers. With compact designs of the laser system and atom cooling system, the mobile atomic gradiometer has the capability to operate out of a lab. By measuring the gravity gradient along a linear path, a demo of detecting an underground tunnel is discussed. No other atomic gravimeters or gradiometers have been reported for detecting underground tunnels. This work is important and would be interesting to the community of quantum sensing and geological surveys. But I think this manuscript is missing some important technical details. I have a few questions and comments for the authors. After they are addressed, I would recommend that this manuscript can be accepted.

1. In Figure 1 (b), why does the upper cloud always have a colder temperature than the upper cloud? Would there be any systematic effects on gradiometry due to the difference between the upper and lower clouds?
2. In Figure 1 (b), what is baseline drift?
3. How big is the quantum gravimeter? What is the physical size of the sensor head? How heavy is it? How big is the laser and electronic system? Can the authors add a picture of the gradiometer when it was operating on top of the tunnel?
4. In figure 1. C, can the authors provide more information about the measurement? What does "Measurement number" mean? How is the error bar calculated? How many are used to calculate the average of each point? How long to measure each data point? How often was the test mass moved? What is the repeatability of the test mass after moves?
5. In the sensor noise budget, the limiting error source is detection noise due to background light, which is about $344 \text{ E}/\sqrt{\text{Hz}}$. But in the insert ellipse in Figure 2, it seems that more data points at the peak and trough, and fewer data points in the middle area, which may indicate that it is limited by phase noises. Can the authors explain this?
6. Is this tunnel particularly constructed for a test measurement? Can the authors add more information about the tunnel? Where is the tunnel? What is the terrain around the tunnel?
7. Because the gradiometer measures gravity at two locations with a height difference of about 1 meter. Would the tidal effect have a different impact on the upper or lower gravimeter? Is the tidal effect corrected?
8. I would recommend that the authors should provide more details in the methods, such as the laser power for MOT or Raman, laser beam size, size or shape (material or coating) of the in-vacuum mirror, MOT coils, and compensation coils.
9. Regarding the tilt alignment, how is the inclinometer attached to the gradiometer?
10. In extended data figure 1, do the going-up beam and going-down beam contain the same laser frequency and power? Can the authors give more details about how the laser beams, frequency,

and power are switched or controller during an interferometry sequence?

Referee #2 (Remarks to the Author):

Dear authors.

It was for me a pleasure to read your manuscript, which describes a very important and novel development in the area of gravity field measurements. I suggest its direct publication (with very small corrections).

In my review I will try to describe my standpoints to all important aspects of the evaluation of your manuscript:

A) Summary of the key results:

Authors describe the construction and application of a new quantum gravity gradient instrument. Results are sound and important from the scope of theoretical and applied gravimetry.

B) Originality and significance:

From my personal point the presented results are of great importance in the area of modern and novel approaches to the of gravity field measurements. I can imagine a great spectrum of applications, when such a measuring instrument would be accessible to academic institutions and also industry. The idea of the vertical gradient computation is new in quantum gravimetry and will help to reduce outer influences to measured values and improve the its precision. This idea is used also in magnetometry, when fluxgate sensors are used for the measurements of vertical gradient of magnetic induction.

C) Data & methodology:

Described new approach is very well presented - its principle and also one example of its application. For me personally, very interesting is the method of the quantitative description of the source of detected anomaly (from the tunnel), which is base on the probability approach. It is very original and differs from the standard inversion procedures, which are usually used in applied gravimetry. Very illustrative is also the Fig. 2 in the extended data section, where is well presented the role of various parameters on the received solution.

D) Appropriate use of statistics and treatment of uncertainties:

Estimations of the uncertainties of the acquired gravity gradient values is performed in a good way. Estimated value of 20 E for the developed instrument is a very optimistic value - from the scope of its practical applications.

E) Conclusions:

Conclusion is well written in my opinion - I do not have any principal comments to its structure and content.

F) Suggested improvements:

I have few very small comments/suggestions:

- It would be maybe good to evaluate the use of the terms gravity and gravitation in the manuscript. I would suggest to use the term gravity in all cases, where the acquired field is described (also in the title of the paper) and the term gravitation only in the case of Newtonian gravitation law application (e.g. in the term Newton's gravitational constant). I like the new term "gravitational cartography". It is quite unusual (in practice terms like gravity survey or exploration are used), but this new term is very adequate. Also in this case, I would suggest to change it to the term "gravity cartography". But this is just a suggestion, I will leave the decision on authors.
- It would be good to give the precise information about the vertical distance between the two sensors, in which the measurement procedures are performed and the vertical level to in which the acquired value of the gravity gradient should be bound (referenced). Also during the description of the results from the practical measurement on the road, the vertical distance of the sensors from the road is important (for applying of forward modelling procedure).

G) References:

In my opinion, all relevant and important references are listed. Maybe that when authors would add an information about the very time consuming procedure in measuring of vertical gravity gradients by classical relative spring gravity meters - the following paper could be added in the references list:
Zahorec P., Papco J., Mikolaj M., Pasteka R., Szalaiova V.:
The role of near topography and building effects in vertical gravity gradients approximation. First Break, Vol.32, Iss.1, 65-71,
<https://doi.org/10.3997/1365-2397.2013033>

H) Clarity and context:

Manuscript is written in a very good style, abstract is well conceived and gives clear summary of this contribution. Introduction and Conclusion parts are also well, I do not have any principal comments. Few small comments/suggestion I have listed in point F of this review.

I wish to the authors good luck in their future scientific developments.

With best regards
Roman Pasteka

Referee #3 (Remarks to the Author):

The manuscript by Stray et al. shows for the first time the use of a quantum sensor for detailed cartography measurements. In particular, the authors focused their attention to a non-invasive precise determination of underground features through the measurement of their weak gravitational signals. For this, they developed a gravity gradiometer based on atom interferometry, showing an absolute sensitivity of 20 E in an averaging time of about 10^3 s.

The article is well written and seems clearly understandable also by a broad audience, so, not only for specialists in atom interferometry techniques and methods.

The realization of portable atom interferometers and the use in field measurement is a hot topic and the experimental results obtained with this innovative "hourglass" configuration is of high level and represents step forward in the field, also showing better performances with respect to already existing commercial devices. Moreover, the bibliography gives appropriate credit to earlier work.

However, I have a couple of comments on the manuscript concerning both the techniques and the experimental results. I list them here:

- line 55+extended data figure 1. While it is clear the advantage of the hourglass configuration which helps in the rejection of common mode noise acquired by the two separated interferometer, due to a high degree of symmetry in cold clouds production and atom's trajectories for the upper and lower interferometer, no detailed information is given on the laser beam configuration used in the production and in the interferometer sequence. In particular, the two corner cubes inserted in the vacuum system used for generating the retroreflected MOT beams (via the use of a single optical beam), effectively reduce the optical aperture for the Raman beams employed in the interferometer.

I could not find detailed information on the dimension of the Raman beams with respect to this aperture and no mention is given to a possible systematic effect due to diffraction of the Raman beams given by the reduced aperture.

It is highly probable that at this sensitivity level the effect might be negligible, but this might become a limitation in planning future experiment implementing large momentum transfer techniques.

I'd like the author consider adding more detailed information on the actual dimension of the free space inside the tube vs. the diameter of the Raman beams.

Moreover, are the Raman beams coupled in the same fibres and collimators used for the MOT? In this case the symmetry given by "hourglass" configuration is broken. While both atomic clouds fall downwards (from the picture about 20-25 cm) before detection, only the lower one will experience extra light field from the reflection from the bottom MOT mirrors. Although this is off resonant light, it might induce a differential phase shifts due to AC Stark effect. Would the author clarify the effect of this asymmetry in the system in terms of projected limitation to the total uncertainty in gravity gradient determination or take it into account in the estimation of systematic effects?

- line 119-120. In view of a future implementation in real field measurements of such a new device, it might be important to clarify here which is the minimum set of additional preliminary measurements ("auxiliary data" in the manuscript) that are requested to perform a precise determination of underground features, as shown here.

Indeed, it is not clear how the accuracy of these pre-measurement (not declared in the manuscript) affects the total uncertainty of determination of the underground feature as for example in absolute position and density. Has the accuracy of the pre-measurement been considered in the determination of the confidence level for position and density determination? Will a better accuracy on pre-measurement help the determination with the atom interferometer?

- line 143 figure 3d). Not clear what this graph is showing. The vertical axis comes with no label, (it must be a probability distribution). Please clarify. The same applies to Extended fig.3 on "Method" section.

- The authors declare an uncertainty of 20 E while there are several sources of systematic corrections not only in the measurement of the underground survey, which are larger than this

value, but also in the absolute calibration of the gradiometer itself.

Indeed, it is evident that this value (20 E) comes directly from the Allan Deviation shown in fig. 2 which is basically a measurement of the statistical uncertainty only. So, when the authors use the word "uncertainty" are they referring to "statistical uncertainty" only? Are they referring to the quadrature sum of the statistical and systematic error on the gradient determination, instead? Please clarify. In this case you may call it "total uncertainty", or indeed you may consider using the term "accuracy".

-Connected to the previous comment, is the sensitivity of the instrument constant as a function of the value of the measured gradient? As the gradient changes, the ellipse data coming from the two simultaneous interferometry will change orientation and the sensitivity in the ellipse fitting will change. In which specific conditions have been taken the measurement shown in fig.2?

-line 194. While the sensitivity of the instrument is fully characterized, no information is given on the measurement range of the instrument. This assumes particular importance for outdoor measurements in which large gradient might also be expected.

Moreover, as the sensitivity increases for example with the implementation of large momentum transfer techniques (up to a factor of 100 improvement in sensitivity, as the author mention), I expect the overall measurement range to be reduced by the same quantity. Is there a way to overcome this possible limitation in field measurement? This would result important especially in cases where previous knowledge of the gradient or "auxiliary data" on mass distribution is not known. Please comment on that.

In conclusion, the manuscript is well written and clear, the results presented are novel and very interesting for a broad audience. However, I recommend the authors consider the above comments and improve the manuscript with additional information and details on the device, particularly focusing to readers outside the field of atom interferometry to help understanding the potentiality of these new quantum sensors.

Author Rebuttals to Initial Comments:

Referee #1 (Remarks to the Author):

The authors demonstrated a mobile atomic gradiometer for detecting underground tunnels. The gradiometer is constructed with two vertically separated atom interferometers. With compact designs of the laser system and atom cooling system, the mobile atomic gradiometer has the capability to operate out of a lab. By measuring the gravity gradient along a linear path, a demo of detecting an underground tunnel is discussed. No other atomic gravimeters or gradiometers have been reported for detecting underground tunnels. This work is important and would be interesting to the community of quantum sensing and geological surveys. But I think this manuscript is missing some important technical details. I have a few questions and comments for the authors. After they are addressed, I would recommend that this manuscript can be accepted.

Thank you for the positive comments about the work and for providing detailed feedback in making the manuscript more accessible to the reader. We have added all the requested details, as detailed against each suggestion below.

1. In Figure 1 (b), why does the upper cloud always have a colder temperature than the upper cloud?

Would there be any systematic effects on gradiometry due to the difference between the upper and lower clouds?

Thank you for these questions.

We believe the difference in temperature is predominantly due to differences in residual magnetic fields limiting the achievable temperature during polarisation gradient cooling. These are present due to the specific design of the magnetic shield that we use, which has an entry point for cabling in the lower shield. We believe there is also a difference in the alignment of the cooling light for the two chambers, and this may be relevant. The temperature of the lower cloud can in principle be lowered using compensation coils, but in optimising the sensor we have found that the scheme used here provided the best sensor performance.

To address this comment, we have added the following statement in the methods section after mentioning the temperatures – *‘The differences in temperature arise from differences in local residual magnetic fields, arising primarily due to the magnetic shield geometry, and small differences in optical alignment.’*

A difference in temperature can cause systematic effects for gradiometry. The most relevant of these is the AC Stark shift, due to differences in cloud size over time. This will provide a light intensity dependent offset on the measured gravity gradient. It is not directly possible for us to determine if the difference in temperature is a dominant contribution to any AC Stark shifts, as that could also arise due to small differences in the position of the two clouds within the beam, and differences in the laser beam profile inside the vacuum system. We stabilise the light intensity in order to reduce changes in these offsets over time. We also address drift in the gravity gradiometer output through the procedure described in the methods section.

To provide the reader with some insight on this, we have added the following text to the part of the methods section which discusses drift: *‘The leading source of drift is believed to be due to the AC Stark shift, with this also being relevant due to the difference in the temperature of the two clouds.’*

We have also added the following statement into the methods section, at the end of the paragraph describing the laser system: *‘The intensity of the Raman beams is actively stabilised using feedback from a photodiode to control acousto-optic modulators, which are also used to perform the laser pulses.’*

2. In Figure 1 (b), what is baseline drift?

Thank you for this comment. Here we were referring to the change in the separation of the two clouds of the gradiometer over time.

We have made this clearer by amending the caption of Figure 1b to instead refer to *‘the relative change of the 1 m cloud separation baseline over time’* **and also changing the figure axis label to** *‘Change in baseline (ppm)’*.

We have also amended the reference in the text to now state *'The resulting changes in baseline are below 50 ppm'* rather than referring to baseline stability.

Similarly, in the methods extended data table 1 we have changed the relevant entry such that it refers to *'Changes in cloud separation baseline'*.

3. How big is the quantum gravimeter? What is the physical size of the sensor head? How heavy is it? How big is the laser and electronic system? Can the authors add a picture of the gradiometer when it was operating on top of the tunnel?

Thank you for this comment.

We have addressed this by including a new extended data figure containing a photograph of the gradiometer above the tunnel, with a caption including all of the requested details. The caption is as follows:

'Showing the gravity gradiometer on the survey line, above the tunnel. The main sensor head (blue cylinder) stands at 1.87 m high, varying by 5 cm depending on the setting of the adjustable feet. This places the measurement positions of the sensor at approximately 0.5 m from the road surface for the lower sensor, and 1.5 m for the upper sensor. The floor footprint of the sensor head is 0.64 m by 0.6 m, and the upright cylinder has a diameter of 0.27 m. The total weight of the sensor head is approximately 75 kg. This is connected to a flight case, which contains the lasers and control system. This has an internal height of 24 U standard rack units (1 U = 4.4 cm), an external footprint of 1.10 m by 0.46 m and height of 1.34 m. A secondary case is placed on top of this, with dimensions of 0.50 m by 0.59 m by 0.46 m. The combined weight of these cases is approximately 250 kg. The system operates on a single mains wall socket, drawing approximately 800 W and having a short-term battery holder over. It can also operate on a generator supply without any observed additional noise. Also shown is a frame used to move the sensor head, and prism used to reference position with a total station.'

Due to this, we have also amended the labelling of the remaining figures.

4. In figure 1. C, can the authors provide more information about the measurement? What does "Measurement number" mean? How is the error bar calculated? How many are used to calculate the average of each point? How long to measure each data point? How often was the test mass moved? What is the repeatability of the test mass after moves?

Thank you for this comment, and apologies that this sub-figure was not sufficiently explained.

Throughout the measurement, the sensor was left to run continuously and the test mass moved relative to the sensor. The 'Measurement number' is a time sequential number used to indicate the period of time the test mass was in a given position. The test mass starts close to the sensor, and then is moved away. Each time the mass is moved, the measurement number increases by 1.

Through this, odd measurement numbers indicate the test mass being close to the sensor and even measurement numbers indicate it is moved away from the sensor.

The data points are each formed from an average of the gravity gradient that is extracted from 8 ellipses. Each ellipse contains 25 shots (independent runs of the atom interferometer), with each shot taking 1.5 s and the whole ellipse taking 38 s. This is the same procedure as described in the Methods section for other measurements of the manuscript (apologies that it was not made clear that this was used here also). The error bar is the standard error of the gravity gradient extracted from the 8 ellipses.

The time spent with the mass in each position was between 17 and 24 minutes. The repeatability of the test mass position was below 1 cm, through use of a mechanical stop. There was no variation in the mass or density of the test mass.

We have addressed this in the manuscript by amending the caption for Fig. 1c to the following: *'Measurement of the gravity gradient variation caused by movement of a test mass between two positions – either close to the sensor (open points) or displaced from the sensor (solid points). Each measurement number represents a specific position of the test mass, with the odd measurement numbers having the mass close to the sensor. Each data point is formed from the average of eight gravity gradient measurements, with each of those containing 25 shots from the atom interferometer each taking 1.5 s. The error bar for each data point is the standard error of the eight gravity gradient readings. The test mass was moved approximately every 20 minutes, with a variation of ± 3.5 minutes, and its position was repeatable to approximately 1 cm. The modelled projection of the change in gravity gradient signal, ΔG_{zz} , is shown in red.'*

5. In the sensor noise budget, the limiting error source is detection noise due to background light, which is about $344 \text{ E}/\sqrt{\text{Hz}}$. But in the insert ellipse in Figure 2, it seems that more data points at the peak and trough, and fewer data points in the middle area, which may indicate that it is limited by phase noises. Can the authors explain this?

Thank you for the question and observation. We believe this is a natural consequence of the way our ellipse is formed. The shape of the ellipse is determined by the difference between the two interferometer phases, and the positions of the individual points around the ellipse are determined by phase shifts that are common to the two interferometers, arising from common mode vibration, for example (they are also perturbed by noise, of course, but that will be ignored here). In addition to external vibration, we also deliberately impose a random phase shift on to the interferometers in order to ensure that the points are spread around the entire ellipse: this ensures that the ellipse-fitting algorithm operates well. These random phases are selected from a uniform distribution between 0 and 2π ; thus, on average, the common phase shifts are uniformly distributed, i.e. all phase values are equally likely. However, this does not result in a uniform density of points around the perimeter of the ellipse, except in the special case in which the phase difference is $\pi/2$ and the ellipse becomes a circle. We show this below with two simulated ellipses with common phase shifts that are equally spaced. It can be seen that in the second case, which uses a 0.88 radian phase shift to give a similar ellipse shape to that shown in Fig. 2 inset, there is a clustering at the extremal points of the ellipse, corresponding to what is seen in the experimental data.

We have addressed this comment in the manuscript by adding a note at the end of the caption of Fig. 2: *'The deviation of the ellipse from a circle gives rise to a clustering of points around the extremal points of the ellipse.'*

We have also added the following into the methods section, when talking about the ellipse formation (end of paragraph 6 in the revised version): *'In addition to random noise arising due to vibration, we add a deliberate random phase value, from between 0 and 2π , to the final pulse of the interferometer. At ellipse phases that do not correspond to a circle, a clustering of points around the extremal points of the ellipse is visible even for uniform noise.'*

6. Is this tunnel particularly constructed for a test measurement? Can the authors add more information about the tunnel? Where is the tunnel? What is the terrain around the tunnel?

Thank you for these questions, and apologies for not previously including these details. No, the tunnel was not constructed for the measurement. It is a pre-existing and functional multi-utility tunnel. The tunnel is on the University of Birmingham campus. The tunnel is beneath a working private road way, which is closed at one end. The central axis of the tunnel is roughly aligned with the road. The road is used by cars for parking, and for deliveries to the workshop and such activities continued during the trial. It is between a large auditorium (450 persons, hosting musical events) and a large unheated workshop building, in which materials are stored and machining takes place. The specific measurement site is roughly 50 metres from our main mechanical workshop. The survey line is over the road, between these two buildings. The two buildings are roughly 15 metres apart. A schematic representation of the buildings (from their building plans) is shown in Figure 3b.

We have addressed this comment in the manuscript by amending the paragraph introducing the tunnel, to provide more information regarding these points, as follows: *'To demonstrate the potential for gravity cartography, a 0.5 m spatial resolution survey was performed along an 8.5 m survey line above a pre-existing multi-utility tunnel. This is a 2 m by 2 m internal cross-section tunnel with a reinforced concrete wall of approximately 0.2 m thickness. It is situated underneath a road surface that is located between two multi-storey buildings. Nearby buildings and terrain around the survey site provide additional signals that can mask targets of interest.'*

We have also more clearly mentioned in the text that the buildings are shown in the figure, and made reference to this also being informed by *'building plans (CAD drawings)'*.

7. Because the gradiometer measures gravity at two locations with a height difference of about 1 meter. Would the tidal effect have a different impact on the upper or lower gravimeter? Is the tidal effect corrected?

Thank you for this question. The tidal effect does have a weak height dependence. For the 1 metre separation of the clouds, we calculate a difference that corresponds to 0.3 milliEotvos using the Hartman Wenzel Catalogue. Due to the resulting similarity of the tidal shift on both gravimeters, the effect of tides is reduced such that it is negligible upon taking a differential measurement and when comparing to other measurement errors. We have therefore not corrected for the tidal effect.

We have addressed this by adding the following text to the methods section (end of paragraph 2 of 'Survey practice and processing of the measurement data'): *'The data, as shown in Fig. 3a, is not corrected for terrain or effects such as tides. Tidal effects are not corrected, being negligible through the differential measurement of the gravity gradient.'*

8. I would recommend that the authors should provide more details in the methods, such as the laser power for MOT or Raman, laser beam size, size or shape (material or coating) of the in-vacuum mirror, MOT coils, and compensation coils.

Thank you for this recommendation.

We have addressed this by adding detail on the laser power for the MOT and Raman beam, their beam waists, and details of the in-vacuum mirrors into the methods section, by expanding the details in the third paragraph (making this also a separate paragraph): *'The light is delivered to the sensor head using polarisation-maintaining optical fibres, with separate fibres for the cooling and Raman beams. These fibres deliver the light to optical telescopes that collimate the light at the desired beam size. The cooling beams have a waist of 24 mm, and contain a typical maximum power of 130 mW. These impinge upon the in-vacuum mirrors, which are 15 mm right-angle prisms (Thorlabs, MRA15-E03), to deliver the horizontal cooling beams. The mirrors are mounted to a titanium structure (attached using Epo-Tek H21D adhesive) in a cross configuration such that there is a 15 mm aperture in their centre. The central portion of the cooling beams pass through these apertures to provide the sixth beam required for the opposite magneto-optical trap. The Raman beams are overlapped with the cooling beams using a polarising beam splitter cube, such that they are then delivered along the same beam axis as the cooling light. The Raman beams, each containing a typical maximum power of 300 mW, have their waist set to 6.2 mm to limit aperturing and diffraction on the central aperture of the in-vacuum mirrors, allowing the Raman beams to pass through the system without being redirected by the prisms.'*

Furthermore, we have added the following detail on the MOT coils and additional coils into the third paragraph of the methods section: *'Each MOT region has two coils each formed of 92 turns of 1 mm Kapton-coated copper wire wound around an aluminium former (fixed using epoxy), with a slit to*

prevent eddy currents. The coils have a radius of 43 mm and separation of 56 mm, to produce a linear gradient field of 12.5 G cm^{-1} at a driving current of 2.5 A. These are located around the vacuum system, such that the strong magnetic field axis of their quadrupole field is along the direction of travel of the cooling beam axis. In addition, two sets of rectangular coil pairs, each having 20 turns, are located around the MOT regions. These have a separation of 100 mm, and dimensions of 320 mm in the vertical and 90 mm in the horizontal, and can be used to compensate residual magnetic fields, or apply offsets. In practice, no compensation fields are used for the molasses phase. In the lower chamber, one coil pair is used to apply a 0.63 G field to adjust the atom cloud horizontal position by approximately 0.5 mm in the MOT phase, improving the interferometer contrast.'

We have also added the following remark after introducing the bias coils: '*with other coils being switched off after the magneto-optical trapping phase.*' to aid some readers.

9. Regarding the tilt alignment, how is the inclinometer attached to the gradiometer?

The inclinometer is attached to the mounting structure of the instrument, to maintain a constant tilt between survey positions. This is fixed to a rigid base plate which forms the main interface between the instrument feet and the rest of the instrument.

We have added the following text to clarify this, into Extended data table 1: '*The inclinometer is used to continuously monitor tilt, and is permanently attached to a rigid base plate which also acts as the main interface between the instrument feet and the rest of the instrument.*'

10. In extended data figure 1, do the going-up beam and going-down beam contain the same laser frequency and power? Can the authors give more details about how the laser beams, frequency, and power are switched or controller during an interferometry sequence?

Apologies that the text was not clear. The going-up and going-down Raman beams do not contain the same frequency and power. In each measurement, the Raman beam configuration is either in 'Mode 1' or 'Mode 2' as shown in the figure. The instrument flips between these using the radio-frequency switch shown in the diagram, in order to operate the wave-vector reversal process.

We have clarified this by amending the relevant part of the figure caption to: '*The Raman transitions are realised using electro-optic modulators (EOMs) to create sidebands at a frequency difference equal to the hyperfine ground state splitting. In contrast to previous approaches, each input direction contains a separate EOM with the driving frequency being only applied to one input direction, such that the laser frequencies for the upward and downward Raman beams are in either Mode 1 or Mode 2 of the spectral configurations shown in the figure. This removes the effect of parasitic Raman transitions that create offsets and contrast loss in conventional modulation based approaches. The use of this laser scheme is enabled through the hourglass configuration allowing independent delivery of the Raman beams, while suppressing phase noise through differential operation. Switching between these two modes changes the input direction of the modulated beam spectrum, changing the direction of the first momentum kick in the interferometer and causing it to open in the opposite direction (dashed lines in the interferometer sequence). This allows a practical implementation of the wave-vector reversal procedure, where the contributions to the phase due to*

the gravitational acceleration are sensitive to the direction of the recoil imparted by the light, while many other effects such as those due to magnetic fields are not.'

Furthermore, the paragraph and details added in response to your comment 8 provide additional related information.

Referee #2 (Remarks to the Author):

Dear authors.

It was for me a pleasure to read your manuscript, which describes a very important and novel development in the area of gravity field measurements. I suggest its direct publication (with very small corrections).

Thank you for the positive comments about the work and for providing detailed feedback to help improve the manuscript and make it more useful to the reader. We have added the requested details and amendments against all of the points, as detailed against each suggestion below.

In my review I will try to describe my standpoints to all important aspects of the evaluation of your manuscript:

A) Summary of the key results:

Authors describe the construction and application of a new quantum gravity gradient instrument. Results are sound and important from the scope of theoretical and applied gravimetry.

B) Originality and significance:

From my personal point the presented results are of great importance in the area of modern and novel approaches to the of gravity field measurements. I can imagine a great spectrum of applications, when such a measuring instrument would be accessible to academic institutions and also industry. The idea of the vertical gradient computation is new in quantum gravimetry and will help to reduce outer influences to measured values and improve the its precision. This idea is used also in magnetometry, when fluxgate sensors are used for the measurements of vertical gradient of magnetic induction.

C) Data & methodology:

Described new approach is very well presented - its principle and also one example of its application. For me personally, very interesting is the method of the quantitative description of the source of detected anomaly (from the tunnel), which is base on the probability approach. It is very original and

differs from the standard inversion procedures, which are usually used in applied gravimetry. Very illustrative is also the Fig. 2 in the extended data section, where is well presented the role of various parameters on the received solution.

D) Appropriate use of statistics and treatment of uncertainties: Estimations of the uncertainties of the acquired gravity gradient values is performed in a good way. Estimated value of 20 E for the developed instrument is a very optimistic value - from the scope of its practical applications.

E) Conclusions:
Conclusion is well written in my opinion - I do not have any principal comments to its structure and content.

F) Suggested improvements:
I have few very small comments/suggestions:
- It would be maybe good to evaluate the use of the terms gravity and gravitation in the manuscript. I would suggest to use the term gravity in all cases, where the acquired field is described (also in the title of the paper) and the term gravitation only in the case of Newtonian gravitation law application (e.g. in the term Newton's gravitational constant). I like the new term "gravitational cartography". It is quite unusual (in practice terms like gravity survey or exploration are used), but this new term is very adequate. Also in this case, I would suggest to change it to the term "gravity cartography". But this is just a suggestion, I will leave the decision on authors.

Thank you for the comment.

We have addressed this by changing the title to refer to 'gravity cartography', and reviewing the text changing appropriate instances to refer to gravity.

- It would be good to give the precise information about the vertical distance between the two sensors, in which the measurement procedures are performed and the vertical level to in which the acquired value of the gravity gradient should be bound (referenced). Also during the description of the results from the practical measurement on the road, the vertical distance of the sensors from the road is important (for applying of forward modelling procedure).

Thank you for this comment.

We have addressed this by making a more direct and clear reference to the separation between the clouds, which provides the vertical distance between the sensors, including the following in the caption of Fig. 1: *'and the relative change of the 1 m cloud separation baseline over time'*.

Furthermore, we have added a more detailed description of the instrument itself into a new extended data figure and included the following into the caption to provide details of the distance to the road surface: *'The main sensor head (blue cylinder) stands at 1.87 m high, varying by 5 cm depending on the setting of the adjustable feet. This places the measurement positions of the sensor at approximately 0.5 m from the road surface for the lower sensor, and 1.5 m for the upper sensor.'*

We have also added this into the survey procedure section of the methods section, as follows: *(with horizontal position being measured using a total station, Leica TS15, and the vertical position from the road surface being approximately 0.5 m for the lower sensor and 1.5 m for the upper sensor)'*

G) References:

In my opinion, all relevant and important references are listed. Maybe that when authors would add an information about the very time consuming procedure in measuring of vertical gravity gradients by classical relative spring gravity meters - the following paper could be added in the references list:
Zahorec P., Papco J., Mikolaj M., Pasteka R., Szalaiova V.:
The role of near topography and building effects in vertical gravity gradients approximation. First Break, Vol.32, Iss.1, 65-71,
<https://doi.org/10.3997/1365-2397.2013033>

Thank you for this suggestion and bringing this helpful reference to our attention.

We have addressed the comment by including this as a new reference 35 (amending the labels of other references accordingly).

H) Clarity and context:

Manuscript is written in a very good style, abstract is well conceived and gives clear summary of this contribution. Introduction and Conclusion parts are also well, I do not have any principal comments. Few small comments/suggestion I have listed in point F of this review.

I wish to the authors good luck in their future scientific developments.

With best regards

Referee #3 (Remarks to the Author):

The manuscript by Stray et al. shows for the first time the use of a quantum sensor for detailed cartography measurements. In particular, the authors focused their attention to a non-invasive precise determination of underground features through the measurement of their weak gravitational signals. For this, they developed a gravity gradiometer based on atom interferometry, showing an absolute sensitivity of 20 E in an averaging time of about 10^3 s .

The article is well written and seems clearly understandable also by a broad audience, so, not only for specialists in atom interferometry techniques and methods.

The realization of portable atom interferometers and the use in field measurement is a hot topic and the experimental results obtained with this innovative "hourglass" configuration is of high level and represents step forward in the field, also showing better performances with respect to already existing commercial devices. Moreover, the bibliography gives appropriate credit to earlier work.

Thank you for the positive comments about the work and for providing detailed feedback to help improve the manuscript and make it more accessible and useful to the reader. We have added details to provide further information or clarification against all of the points, as detailed against each suggestion below.

However, I have a couple of comments on the manuscript concerning both the techniques and the experimental results. I list them here:

- line 55+extended data figure 1. While it is clear the advantage of the hourglass configuration which helps in the rejection of common mode noise acquired by the two separated interferometer, due to a high degree of symmetry in cold clouds production and atom's trajectories for the upper and lower interferometer, no detailed information is given on the laser beam configuration used in the production and in the interferometer sequence. In particular, the two corner cubes inserted in the vacuum system used for generating the retroreflected MOT beams (via the use of a single optical beam), effectively reduce the optical aperture for the Raman beams employed in the interferometer.

I could not find detailed information on the dimension of the Raman beams with respect to this aperture and no mention is given to a possible systematic effect due to diffraction of the Raman beams given by the reduced aperture.

It is highly probable that at this sensitivity level the effect might be negligible, but this might become a limitation in planning future experiment implementing large momentum transfer techniques. I'd like the author consider adding more detailed information on the actual dimension of the free space inside the tube vs. the diameter of the Raman beams.

Moreover, are the Raman beams coupled in the same fibres and collimators used for the MOT? In this case the symmetry given by "hourglass" configuration is broken. While both atomic clouds

fall downwards (from the picture about 20-25 cm) before detection, only the lower one will experience extra light field from the reflection from the bottom MOT mirrors. Although this is off resonant light, it might induce a differential phase shifts due to AC Stark effect. Would the author clarify the effect of this asymmetry in the system in terms of projected limitation to the total uncertainty in gravity gradient determination or take it into account in the estimation of systematic effects?

Thank you for these comments and questions. Apologies that some details were missing here.

We have addressed this by adding the following details on the frequency generation for production, into paragraph 4 of the methods section: *'The light for laser cooling is generated by passing the laser output through an electro-optic modulator (EOM) and generating a sideband at a frequency of approximately 1.2 GHz output from the carrier. This is used to provide a locking signal using the $|F' = 3 \rangle \rightarrow |F' = 4 \rangle$ transition in rubidium-85, placing the carrier frequency such that it is tuneable around resonance with the $|F = 2 \rangle \rightarrow |F' = 3 \rangle$ transition of rubidium-87 to provide the cooling light. A separate EOM is used to provide repumping light resonant with the $|F = 1 \rangle \rightarrow |F' = 2 \rangle$ transition.'*

Furthermore, we have clarified the description of the laser beam configuration in the atom interferometry sequence, amending the text in extended data fig 1 caption: *'each input direction contains a separate EOM with the driving frequency being only applied to one input direction, such that the laser frequencies for the upward and downward Raman beams are in either Mode 1 or Mode 2 of the spectral configurations shown in the figure.'*

Followed by, later in the caption *'Switching between these two modes changes the input direction of the modulated beam spectrum, changing the direction of the first momentum kick in the interferometer and causing it to open in the opposite direction (dashed lines in the interferometer sequence). This allows a practical implementation of the wave-vector reversal procedure, where the contributions...'*

We have provided more input on the beam sizes, their delivery, and the in-vacuum mirrors, we have added the following text to the methods section, making a new fifth paragraph: *'The light is delivered to the sensor head using polarisation-maintaining optical fibres, with separate fibres for the cooling and Raman beams. These fibres deliver the light to optical telescopes that collimate the light at the desired beam size. The cooling beams have a waist of 24 mm, and contain a typical maximum power of 130 mW. These impinge upon the in-vacuum mirrors, which are 15 mm right-angle prisms (Thorlabs, MRA15-E03), to deliver the horizontal cooling beams. The mirrors are mounted to a titanium structure (attached using Epo-Tek H21D adhesive) in a cross configuration such that there is a 15 mm aperture in their centre. The central portion of the cooling beams pass through these apertures to provide the sixth beam required for the opposite magneto-optical trap. The Raman beams are overlapped with the cooling beams using a polarising beam splitter cube, such that they are then delivered along the same beam axis as the cooling light. The Raman beams, each containing a typical maximum power of 300 mW, have their waist set to 6.2 mm to limit aperturing and diffraction on the central aperture of the in-vacuum mirrors, allowing the Raman beams to pass through the system without being redirected by the prisms.'*

We have amended Figure 1a and its caption to indicate that the cooling and atom interferometry beams do not have the same waist, with the caption as follows: *'The beam delivery is indicated with arrows (see Methods for details). The cooling beams (red) are deflected by the in-vacuum mirrors (blue) to provide cooling in all directions, with the central portion of each input beam passing through the aperture between the mirrors to provide the final cooling beam for the opposite magneto-optical trap. The atom interferometry beams (yellow arrows) have a smaller beam waist, such that they pass through the mirror aperture without significant clipping.'*

This also clarifies that the Raman beam is smaller than the aperture of the mirrors, meaning we do not observe effects due to diffraction in the current system. Furthermore, this means that there is not an extra light field for the lower interferometer. We have clarified the Raman beam size compared to the mirror aperture, in the above text.

To address this point, we have added the following additional text following on from where the previous amendment to the methods finished: *'While limiting aperturing on the mirrors in the current instrument, it may be desirable to use a larger Raman beam than the aperture in more compact systems or those aiming to further reduce laser beam inhomogeneity induced dephasing. Diffraction from the aperture would need to be given due consideration if pursuing this, as would the potential for additional light shifts due to in this case one interferometer seeing additional light fields from mirror reflections.'*

- line 119-120. In view of a future implementation in real field measurements of such a new device, it might be important to clarify here which is the minimum set of additional preliminary measurements ("auxiliary data" in the manuscript) that are requested to perform a precise determination of underground features, as shown here.

Indeed, it is not clear how the accuracy of these pre-measurement (not declared in the manuscript) affects the total uncertainty of determination of the underground feature as for example in absolute position and density. Has the accuracy of the pre-measurement been considered in the determination of the confidence level for position and density determination? Will a better accuracy on pre-measurement help the determination with the atom interferometer?

Thank you for this question. One of the advantages of inference process we use is that we do not need detailed auxiliary data in order to determine the position information of the tunnel. The inference of the position data, i.e. that shown in Figure 3c, is determined solely with the atom interferometry data (the gravity gradient and the spatial positions it is measured at, with the latter being obtained with a total station) and knowledge of typical value ranges of various parameters (those provided in Extended data Table 2, which also shows the ranges used). For instance, we assume that the soil density is within the expected range for the type of soil at the survey site ($1800 \pm 100 \text{ kg/m}^3$). This is typical for what a user would experience in a real survey. In what we show in the paper, there is not any use of auxiliary data to refine these assumptions (prior distributions). Furthermore, for clarity, there is no use of auxiliary data in correcting the atom interferometer data shown in Figure 3a. If we were to use auxiliary data to better define the prior distributions, this would, as for any inference process, result in narrower posterior distributions and therefore better estimates of parameters such as position.

The primary use of auxiliary data in the work is to estimate of the expected local gravity gradient signal from a 'site model' (dashed line in Figure 3a). This is for comparison only, and not used in the inference. We also use topographical information when switching the inference process to make estimates of soil density by assuming the tunnel geometry is known, i.e. to realise the results shown in Figure 3d. In this case, the topographical information is needed. The accuracy of the density estimation is limited by the atom interferometer data accuracy.

We have addressed this comment by clarifying this part of the text: *'Nearby buildings and terrain around the survey site provide additional signals that can mask targets of interest. To estimate the expected signal from the tunnel, a model of the site was constructed using an air/soil contrast infinite cuboid void, taking into account local buildings and terrain. The parameters for the model were informed using building plans (CAD files), with these being cross-checked using ground penetrating radar, and auxiliary data from on-site measurements such as topography scanning. This provided an estimate peak signal from the tunnel of 150 E, which corresponds to a phase change of 17.5 mrad for the atom interferometer. Fig. 3a shows a comparison between the site model and the atom interferometer data, showing that the measurement data is consistent with what is expected for a gravity gradient anomaly with the expected location and size of the tunnel. A scale representation of the site and tunnel, including local buildings and site topology, is shown in Fig. 3b.'*

Furthermore, we have clarified the following paragraph, and made more explicit reference to the information that is used in the inference process: *'For this purpose, we have developed a Bayesian inference method and applied this to the gradiometer data with a data-generated model of a buried cuboid assumed a priori. This uses the gradiometer data in conjunction with estimates of the site and geophysical parameters (as detailed in Extended data Table 2 of the Methods) to make quantitative predictions of the depth and spatial extent of the anomaly. For instance, we assume that the soil density is within the expected range for the type of soil at the survey site by using a Gaussian distribution, with a mean of -1.80 g/cm^3 , to represent a void in surrounding soil, and standard deviation of 0.10 g/cm^3 .'*

We have also clarified this for the switched inference of the soil density: *'Furthermore, by assuming a priori knowledge of the tunnel geometry and including topographical information of the survey site, the focus of the inference was switched to infer the soil density (Fig. 3d).'*

We have also amended the end of paragraph 2 of the inference section of the methods, also moving this to the end of the section to prevent ambiguity between the site model and the cuboid model used for the inference: *'The signals arising from local features are used to create a distinct site model. This is used to provide an estimate of the expected shape of the gravity gradient signal over the site, for comparison with the inference output. These features include the tunnel of interest, basements from nearby buildings, walls and a drain. They are shown in the scale drawing of Fig. 3b.'*

- line 143 figure 3d). Not clear what this graph is showing. The vertical axis comes with no label, (it must be a probability distribution). Please clarify. The same applies to Extended fig.3 on "Method" section.

Apologies for this, and thank you for spotting it. They are indeed probability distributions.

We have addressed this comment by adding the axes labels of Fig. 3d and Extended data Fig. 3 (now called Extended data Fig. 4, due to an additional figure) accordingly.

- The authors declare an uncertainty of 20 E while there are several sources of systematic corrections not only in the measurement of the underground survey, which are larger than this value, but also in the absolute calibration of the gradiometer itself.

Indeed, it is evident that this value (20 E) comes directly from the Allan Deviation shown in fig. 2 which is basically a measurement of the statistical uncertainty only. So, when the authors use the word "uncertainty" are they referring to "statistical uncertainty" only? Are they referring to the quadrature sum of the statistical and systematic error on the gradient determination, instead? Please clarify. In this case you may call it "total uncertainty", or indeed you may consider using the term "accuracy".

Thank you for this comment. As you have correctly identified, it is the statistical uncertainty we are referring to with this value.

We have addressed this comment by in all cases (throughout the text, and in figure 4 caption) stating that it is the statistical uncertainty we are referring to in relevant instances.

-Connected to the previous comment, is the sensitivity of the instrument constant as a function of the value of the measured gradient? As the gradient changes, the ellipse data coming from the two simultaneous interferometry will change orientation and the sensitivity in the ellipse fitting will change. In which specific conditions have been taken the measurement shown in fig.2?

Thank you for this comment and observation. The sensitivity of the instrument is a weak function of the measured gradient, as changes in the gradient will indeed change the orientation of the ellipse (i.e. the ellipse phase angle) and therefore fitting sensitivity. We have carried out as-yet-unpublished studies which show that the error in the measured gradient is typically a few parts in a thousand of the change in gradient, so a 100 E change would have an error of order 0.5E, which is much less than the current measurement errors. Note, however, that these errors are deterministic and can be corrected for in principle. Furthermore, changes in the ellipse phase angle are small for a given change in gravity gradient, with a 100 E change in gradient corresponding to an 11.6 mrad change in ellipse angle. The measurement of Fig 2 is performed at the ellipse angle shown in the inset, the ellipse in the inset being a typical sub-set of the data set used for the Allan deviation.

We have addressed this by clarifying the text by adding the following to the part of the caption for Fig. 2 that relates to the inset: *'Inset – a typical sub-set of the ellipse data used for the Allan deviation, showing a Lissajous figure of the output signal of the upper and lower interferometers, which is used to extract the gradiometric phase.'*

Furthermore, we have added the following into the 'Survey practice and data processing section' of the methods: *Errors in the ellipse fitting are sensitive to changes in the ellipse opening angle [Barrett]. On the basis of numerical simulations we estimate this effect to be less than a few parts in one thousand, therefore, a 100 E change would be subject to an error of less than 0.5 E. Such errors are therefore small compared to other errors. Such a change in gradient would correspond to an 11.6*

mrاد change in the ellipse shape.'

With the reference in the above being “Barrett, B. et al. Correlative methods for dual-species quantum tests of the weak equivalence principle. *New J. Phys.*, **17**, 085010, <https://doi.org/10.1088/1367-2630/17/8/085010> (2015)”.

-line 194. While the sensitivity of the instrument is fully characterized, no information is given on the measurement range of the instrument. This assumes particular importance for outdoor measurements in which large gradient might also be expected.

Moreover, as the sensitivity increases for example with the implementation of large momentum transfer techniques (up to a factor of 100 improvement in sensitivity, as the author mention), I expect the overall measurement range to be reduced by the same quantity. Is there a way to overcome this possible limitation in field measurement? This would result important especially in cases where previous knowledge of the gradient or "auxiliary data" on mass distribution is not known. Please comment on that.

Thank you for this helpful comment, which we agree is important to consider when discussing the large momentum transfer techniques – where indeed it will reduce the measurement range. The sensitivity can be increased through two means – reducing noise (such as improving electronic noise, reducing atom shot noise, reducing dephasing in the interferometer) and increasing the phase difference that is accumulated (for example using techniques like large momentum transfer). The former will not reduce the measurement range, while the later will.

As shown in Figure 4a, a tunnel such as this is one of the larger features of interest for the applications we mention. The majority of these we would expect to focus on surveying, and therefore looking for changes in the gravity gradient. From airborne gravity gradiometer data that we are aware of, signals of up to 300 E could be expected. In modelling we have completed for sub-surface carbon capture and storage we expect signals of up to 400 E, but do not see cases where the signal would exceed such values (as shown in Fig. 4a). For the instrument in its current configuration, the change in phase due to the 150 E tunnel signal is 17.5 mrad, which means we would expect 46.6 mrad shifts for the largest features of interest. These are within the 2π radian measurement range of the instrument in its current configuration without special modification. Increasing sensitivity with a 100x increase in large momentum transfer would increase the phase shift caused by a given gravity gradient signal by 100x. In such a configuration, the tunnel would then correspond to a 1.75 rad shift, and the largest shifts we expect would be up to 4.66 rad. This remains within the measurement range, but significantly larger LMT would become an issue.

Large momentum transfer (and other techniques for increasing phase for a given signal) are just one method for increasing performance, and others could include reducing instrumental noise – which does not come with a reduction in measurement range.

To address this point, we have provided further information to the reader by providing the expected phase change due to the tunnel. This is found within the text when discussing the signal size of the

tunnel: *'This provided an estimate peak signal from the tunnel of 150 E, which corresponds to a phase change of 17.5 mrad for the atom interferometer.'*

We have also provided the following regarding the measurement range in the survey practice section of the methods: *'This phase shift can be compared to a 2π measurement range, meaning that measurement range of the instrument in this configuration is relevant to the majority of practical features of interest (these being typically below 400 E).'*

In conclusion, the manuscript is well written and clear, the results presented are novel and very interesting for a broad audience. However, I recommend the authors consider the above comments and improve the manuscript with additional information and details on the device, particularly focusing to readers outside the field of atom interferometry to help understanding the potentiality of these new quantum sensors.

Reviewer Reports on the First Revision:

Referee #1 (Remarks to the Author):

The authors replied my review comments and revised the manuscript. All my questions are addressed. I recommend that this manuscript can be accepted for publication.

Referee #3 (Remarks to the Author):

The authors provided an extensive and complete answer on all the referee's comments. The manuscript (text and figures) have been modified and upgraded following all the referee's advice.

I have no further comment from my side.
The article can be published as is.